# Scalable Decision-Making in Stochastic Environments through Learned Temporal Abstraction

**Baiting Luo[1], Ava Pettet[3], Aron Laszka[2], Abhishek Dubey[1], Ayan Mukhopadhyay[1]**
[1]Vanderbilt University, [2]Pennsylvania State University, [3]Nissan Advanced Technology Center
`{baiting.luo, abhishek.dubey, ayan.mukhopadhyay}@vanderbilt.edu`
`alaszka@psu.edu, ava.pettet@nissan-usa.com`

## Abstract

Sequential decision-making in high-dimensional continuous action spaces, particularly in stochastic environments, faces significant computational challenges. We explore this challenge in the traditional offline RL setting, where an agent must learn how to make decisions based on data collected through a stochastic behavior policy. We present *Latent Macro Action Planner* (L-MAP), which addresses this challenge by learning a set of temporally extended macro-actions through a state-conditional Vector Quantized Variational Autoencoder (VQ-VAE), effectively reducing action dimensionality. L-MAP employs a (separate) learned prior model that acts as a latent transition model and allows efficient sampling of plausible actions. During planning, our approach accounts for stochasticity in both the environment and the behavior policy by using Monte Carlo tree search (MCTS). In offline RL settings, including stochastic continuous control tasks, L-MAP efficiently searches over discrete latent actions to yield high expected returns. Empirical results demonstrate that L-MAP maintains low decision latency despite increased action dimensionality. Notably, across tasks ranging from continuous control with inherently stochastic dynamics to high-dimensional robotic hand manipulation, L-MAP significantly outperforms existing model-based methods and performs on-par with strong model-free actor-critic baselines, highlighting the effectiveness of the proposed approach in planning in complex and stochastic environments with high-dimensional action spaces.

## 1 Introduction

Planning-based reinforcement learning (RL) has achieved remarkable success in domains with discrete, low-dimensional action spaces, such as board games and video games (Silver et al., 2017; Schrittwieser et al., 2020; Ye et al., 2021), and continuous control tasks (Hubert et al., 2021; Schrittwieser et al., 2021). However, extending these methods to high-dimensional continuous action spaces, especially in stochastic environments, presents significant challenges. Many environments are inherently stochastic or appear stochastic to agents with limited capacity to model complex dynamics. For example, in autonomous driving, the behavior of other vehicles and pedestrians introduces substantial uncertainty that must be processed in real time (Carvalho et al., 2014; Luo et al., 2023a). Recent planning-based offline RL approaches like Trajectory Transformer (Janner et al., 2021) face significant latency issues when trying to model and respond to these stochastic behaviors (Li et al., 2023). Similarly, in robotic manipulation, sensor noise introduces randomness that requires fast adaptive responses (Yang et al., 2023). In such cases, deterministic models often fail to capture the necessary randomness and intricacies (Antonoglou et al., 2022). Moreover, the vast and uncountable nature of continuous action spaces makes traditional planning approaches inefficient when operating directly in the raw action space (Jiang et al., 2023). These inefficiencies are further exacerbated in stochastic settings, leading to "large and long" planning problems where agents must manage numerous continuous variables over extended time horizons. This results in the *curse of dimensionality* and the *curse of history*, significantly hindering effective decision-making (Hubert et al., 2021).

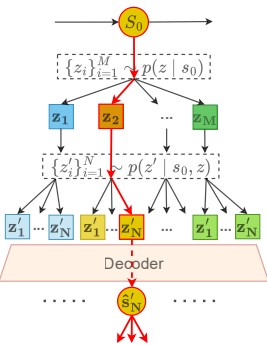 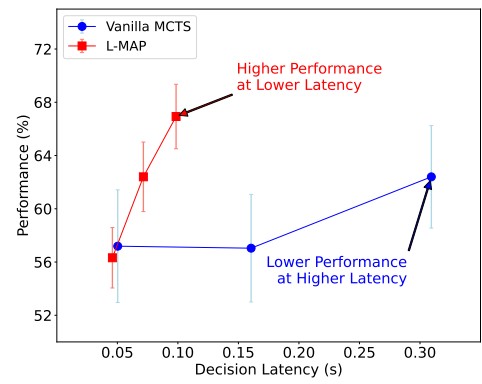

(a) Planning with Pre-constructing search space

(b) Decision Latency vs. Performance

Figure 1: (a) Overview of planning over the pre-constructed search space. (b) As the number of MCTS iterations increases (10, 50, 100 from left to right), using a pre-constructed search space with MCTS achieves better performance with lower decision latency.

In this paper, we posit that planning in such challenging settings could greatly benefit from temporal abstractions, i.e., representations of multi-step primitive behaviors such as macro actions (Dieterich, 2000; Sutton et al., 1999; Barto & Mahadevan, 2003). By leveraging these abstractions, planners can navigate high-dimensional continuous action spaces more efficiently, potentially mitigating the curse of dimensionality and reducing decision-making latency in stochastic environments. This paper considers the standard setting where an agent can access a set of trajectories (i.e., a sequence of state, action, and reward traces) collected through a fixed behavior policy. Given this setting, we propose the *Latent Macro Action Planner* (L-MAP), which constructs a lower dimensional representation of temporally extended primitive actions by using a state-conditioned Vector Quantized Variational AutoEncoder (VQ-VAE) (van den Oord et al., 2017). The encoder integrates the current state and macro-action to generate a discrete latent code. Subsequently, a sequential model (in our case, a Transformer) is employed to autoregressively model the distribution of these latent codes, conditioned on the current state (and the behavior policy). This Transformer facilitates a two-step inference process: initially, given a state, it enables the sampling of probable latent macro-actions under the behavior policy, effectively acting as a prior policy. Subsequently, conditioned on both the state and the sampled macro-action, it generates subsequent latent codes that encapsulate information about expected returns and potential next states. This dual functionality of the Transformer enables efficient exploration of promising action trajectories while forming a compact representation of the plausible trajectories during planning.

As shown in Fig.1a, leveraging these models, we build a latent search space that serves as a structured initialization for planning, encapsulating likely trajectories based on the learned environment dynamics. To address stochasticity and optimize decision-making, we integrate Monte Carlo Tree Search (MCTS) with progressive widening to efficiently navigate this latent space. Initially, the search concentrates on the prebuilt latent space, facilitating rapid decision-making grounded in learned abstractions. If additional computation time becomes available, we progressively widen the search tree to extend the search beyond the prebuilt latent space incrementally. This dynamic expansion strategy enables our method to balance rapid planning using learned abstractions with more exhaustive exploration when computational resources permit. As shown in Fig.1b, this strategy achieves better performance with lower decision latency compared to planning with vanilla MCTS. Upon selecting a latent macro-action, we operate in a *polling control* mode (He et al., 2011; Gabor et al., 2019) wherein MCTS returns only the first primitive action of the recommended macro-action. This approach allows for recovery from locally suboptimal decisions by performing planning at each time step.

We evaluate L-MAP extensively in the offline RL setting across a diverse range of tasks. In stochastic MuJoCo environments (Rigter et al., 2023), L-MAP consistently outperforms both model-based baselines like Trajectory Transformer (TT) (Janner et al., 2021) and Trajectory Autoencoding Planner (TAP) (Jiang et al., 2023), as well as model-free methods such as Conservative Q-Learning (CQL) (Kumar et al., 2020) and Implicit Q-Learning (IQL) (Kostrikov et al., 2022). This demon-

strates L-MAP's robust capability in handling stochastic dynamics. For deterministic MuJoCo tasks, L-MAP shows comparable or superior performance to these baselines, highlighting that our planning approach effectively accounts for stochasticity in the behavior policy, leading to competitive performance even in deterministic environments. Notably, L-MAP scales effectively to high-dimensional tasks, as evidenced by its strong performance on the challenging Adroit hand manipulation tasks. Furthermore, L-MAP's use of temporal abstraction enables lower latency decision-making compared to methods like TT. These results underscore L-MAP's versatility and effectiveness across various types of control problems, from stochastic to deterministic environments, and from low to high-dimensional action spaces.

## 2 PRELIMINARIES

We consider a continuous state and action space Markov Decision Process (MDP) defined by $\{\mathcal{S}, \mathcal{A}, P, r\}$, where $\mathcal{S} \subseteq \mathbb{R}^n$ is the state space, $\mathcal{A} \subseteq \mathbb{R}^l$ is the action space, $P : \mathcal{S} \times \mathcal{A} \rightarrow \Delta(\mathcal{S})$ is the transition function, and $r : \mathcal{S} \times \mathcal{A} \rightarrow \mathbb{R}$ is the reward function. To manage the complexity of these continuous spaces, we introduce macro actions, which are fixed-length sequences of primitive actions. A macro action $m \in \mathcal{M}$ is defined as $m = \langle a_t, \ldots, a_{t+L-1} \rangle$, where each $a_i \in \mathcal{A}$ and $L$ is the length of the macro action. Our goal is to compute an optimal macro-level policy $\pi^* : \mathcal{S} \rightarrow \mathcal{P}_m$ that maximizes the expected discounted return $\mathbb{E}_\pi [R(s, \pi(s))]$.

**Trajectory Representation:** Consider a trajectory $\tau$ of length $T = \kappa \cdot L$ ($\kappa \in \mathbb{N}^+$), which is composed of a sequence of states $s_t \in \mathcal{S}$, fixed-size macro actions $m_t \in \mathcal{M}$, and corresponding return-to-go estimates $R_t = \sum_{i=t}^{T} \gamma^{i-t} r_i$, formally represented as $\tau = (R_1, s_1, m_1, R_{L+1}, s_{L+1}, m_{L+1}, \ldots, R_{(\kappa-1)L+1}, s_{(\kappa-1)L+1}, m_{(\kappa-1)L+1})$.

## 3 METHOD

Planning in continuous action space is hard and computationally challenging, and full enumeration of all possible actions is infeasible. Discretizing the action space is one way to address this challenge, but in practice, enumerating a large set of discrete actions can also be challenging, particularly for online approaches. Sample-based methods offer an efficient approach for handling large and complex domains. These methods sample a subset of actions rather than exhaustively enumerating all possibilities, reducing computational costs while computing optimal policies or value functions (Hubert et al., 2021). Building on these insights, we propose the Latent Macro Action Planner (L-MAP), which learns temporal abstractions in the form of macro-actions and plans using a latent transition model that serves as both a prior policy and a transition model.

### 3.1 DISCRETIZING STATE-MACRO ACTION SEQUENCES WITH VQ-VAE

A key insight from prior work is that a learned state-conditioned discretization can be used to construct a discretization scheme with relatively few discrete actions while maintaining high granularity (Jiang et al., 2023; Luo et al., 2023b). As shown in Fig.2, our approach leverages a learned state-conditioned discretization to enable planning in a lower-dimensional discrete space. Specifically, our encoder processes sequences of state and macro-actions as input. For example, each token is defined as $x_t = (R_t, s_t, m_t)$ and its subsequent token as $x_{t+L} = (R_{t+L}, s_{t+L}, m_{t+L})$. The encoder function is defined as:

$$f_{\text{enc}} (x_t = (R_t, s_t, m_t), \ x_{t+L} = (R_{t+L}, s_{t+L}, m_{t+L})) = (z_t, z_{t+L}), \quad (1)$$

where the transition chunk size is two, resulting in two latent codes assigned per chunk. To elaborate, the encoder first concatenates the input return-to-go estimates, states, and macro actions into two transition vectors. It then applies a sequence model, in our case, a causal Transformer, producing two latent feature vectors for each chunk of transitions.

In stochastic environments, executing the same macro-action $m$ from state $s$ can yield different returns $R$, introducing variability that complicates the vector quantization in VQ-VAE, i.e., note that using the full token $x_t = (R_t, s_t, m_t)$ directly can result in different latent codes $z$ for identical $(s_t, m_t)$ pairs solely due to differences in $R_t$. This challenge can cause the latent space to become fragmented and reflect return variability more than the underlying structure of available actions.

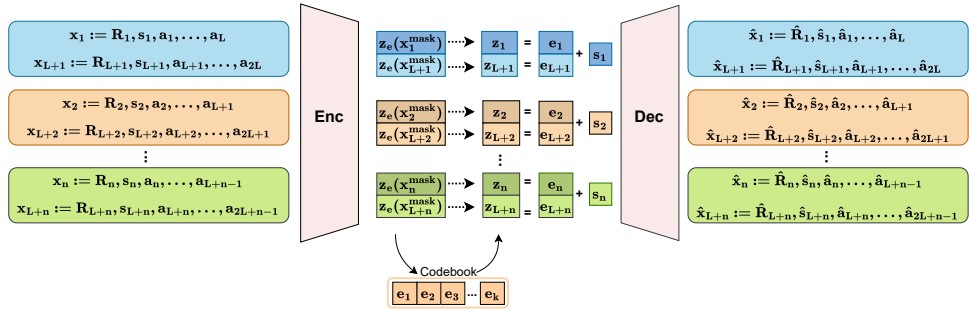

Figure 2: An overview of our VQ-VAE model that discretizes state-macro action sequences

Consequently, the agent might overestimate the returns during decision-making by emphasizing latent codes associated with higher observed returns, neglecting the true distribution of the primitive actions and their expected returns.

To address this issue, we aim to focus the vector quantization process primarily on representations of the state $s$ and macro-actions $m$, while still preserving the ability to reconstruct the return $R$. To tackle this challenge, our approach involves creating two versions of each token $x_t$: the full input $x_t = (R_t, s_t, m_t)$ and a masked version $x_t^{\text{mask}} = (\text{mask}, s_t, m_t)$, where $R_t$ is masked out. The encoder processes both $x_t$ and $x_t^{\text{mask}}$ to generate two embeddings, $z_e(x_t)$ and $z_e(x_t^{\text{mask}})$, respectively. We use $z_e(x_t^{\text{mask}})$ for vector quantization to obtain the quantized latent code $z_q(x_t^{\text{mask}})$. To ensure that the codebook embeddings incorporate information from the full input, including $R_t$, we update the embedding $\mathbf{e}_t$ of the quantized latent code towards $z_e(x_t)$. We modify the loss function by introducing an additional term that encourages the embedding from the masked input to be close to that from the full input. Specifically, we used the embedding from the full input $z_e(x)$ as the learning target for the embedding of the masked input $z_e(x^{\text{mask}})$. The modified loss function is:

$$\mathcal{L} = \log p(x \mid z_q(x^{\text{mask}})) + \|\text{sg}[z_e(x)] - \mathbf{e}\|_2^2 + \beta \|z_e(x^{\text{mask}}) - \text{sg}[\mathbf{e}]\|_2^2 + \|z_e(x^{\text{mask}}) - z_e(x)\|_2^2 \quad (2)$$

where sg denotes the stopgradient operator and the additional term $\left\|z_e(x^{\text{mask}}) - z_e(x)\right\|_2^2$ acts as a regularizer that aligns the embeddings of the masked and full inputs.

Incorporating macro-actions within each token is critical, as it enables the model to capture temporal dependencies across multiple time steps without the need for downsampling. This approach is particularly important in stochastic settings, where downsampling techniques that aggregate states (as in Jiang et al. (2023)) can obscure the stochasticity imposed by the environment's dynamics. The decoder takes the initial state and latent codes as inputs, and outputs the reconstructed trajectories:

$$f_{dec}(s_t, z_t, z_{t+L}) = (\hat{x}_t = (\hat{R}_t, \hat{s}_t, \hat{m}_t), \hat{x}_{t+L} = (\hat{R}_{t+L}, \hat{s}_{t+L}, \hat{m}_{t+L})). \quad (3)$$

The decoding process can be seen as the inverse of the encoding process, except that the initial state $s_t$ is merged into the embeddings of the codes with a linear projection before decoding.

**Latent Transition Model**: Following the discretization process, the subsequent step involves modeling sequences of latent codes in an autoregressive manner using a causal Transformer. The Prior Transformer is conditioned on the initial state $s_t$, achieved by adding the state feature to all token embeddings (Jiang et al., 2023). Primarily, it functions as a transition model in the latent space, enabling the sampling of the next latent code $z_{i+1}$ conditioned on the current code $z_i$ and state $s$. This transition, represented as $T : \mathcal{S} \times \mathcal{Z} \to \mathcal{Z}$, implicitly captures the full $\mathcal{R} \times \mathcal{S} \times \mathcal{M} \to \mathcal{R} \times \mathcal{S} \times \mathcal{M}$ transition in the original space, as each $z$ encodes information about the return-to-go, state and macro-action. Additionally, $p(z \mid s)$ acts as a prior policy for efficient action sampling, allowing rapid selection of probable macro-actions based on learned behaviors from the offline dataset. By operating in the learned latent space, the model potentially reduces computational complexity compared to modeling transitions in the original state-action space, especially for high-dimensional environments. The discrete nature of the latent space allows for efficient sampling, which can be beneficial for downstream tasks such as planning.

## 3.2 PLANNING WITH A LATENT MACRO ACTION MODEL

Planning in high-dimensional environments using learned discrete representations introduces uncertainties from multiple sources. First, the representation learning process introduces uncertainty due to the non-injective mapping from the high-dimensional state-action space to a lower-dimensional latent space. This can result in many-to-one correspondences, where multiple distinct high-dimensional inputs map to the same latent representation, creating apparent stochasticity even in deterministic environments. Second, the environment itself may be inherently stochastic. The detailed analysis of these uncertainty sources is provided in Appendix C.

We argue that taking expectations over latent transitions is beneficial in mitigating all these sources of uncertainty, regardless of whether the environment is deterministic or stochastic. By considering the expected outcomes over multiple latent transitions, we can average out the randomness introduced by the non-injective mapping and inherent stochasticity, leading to more reliable planning decisions. This insight applies broadly to planning methods that employ models with non-injective mapping characteristics. Building on this insight, we employ Monte Carlo Tree Search (MCTS) as our planning algorithm to mitigate the impact of stochasticity arising from non-injective mappings and potential environmental randomness. MCTS iteratively explores the latent space and takes expectations over transitions, allowing for robust planning in the presence of uncertainty.

**Pre-constructing the Latent Search Space.** Our approach leverages a learned latent transition model to generate and evaluate macro actions for planning efficiently. Starting from an initial state $s_0$, we sample $M$ latent codes $z$, each representing a potential macro action. For each sampled latent code $z$, we sample $N$ subsequent latent codes $z'$ to simulate possible future trajectories, capturing the outcomes of these macro actions. We obtain the corresponding state-action transitions and return estimates by decoding these latent pairs $(z, z')$ conditioned on $s$.

To construct the planning tree efficiently, we cache the initial state $s_0$ along with the top-$k$ latent codes $z$ (and their associated information) based on the decoded returns, where $k = \lambda \times M$ and $\lambda \in (0, 1]$ controls the expansion ratio of the tree. The cached latent codes represent the most promising macro actions to consider from the initial state. The latent codes $z'$ are then decoded to obtain a set of reconstructed tokens, i.e., $(R, s, m)$. For each of these states $s$, we sample $B$ latent codes $z''$, representing potential macro actions from $s$ (note that $B$ and $M$ are exogenously defined hyper-parameters). This process is recursively applied, allowing us to expand the planning tree while controlling its growth through the parameter $\lambda$. By focusing on the most promising macro actions at each state, we maintain a compact and informative planning structure that efficiently explores the state-action space at a macro level.

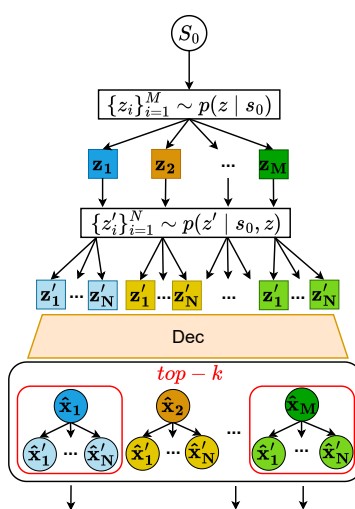

Figure 3: Pre-construction of the latent search space by sampling and evaluating latent macro-action codes, caching the top-k candidates, and recursively expanding the planning tree for efficient macro-level planning.

**Selection.** Starting from the cached tree structure, MCTS iteratively expands and evaluates nodes, allowing for a more comprehensive exploration of the state-action space. For each state $s$ in the tree, MCTS selects one of the top-$k$ cached latent codes $z$ based on the Upper Confidence Bounds for Trees (UCT) (Kocsis & Szepesvári, 2006): $\text{UCT}(s, z) = Q(s, z) + c\sqrt{\frac{\log(N(s))}{N(s,z)}}$ where $Q(s, z)$ represents the value of executing macro action $z$ in state $s$ (estimated through the decoded return-to-go), $N(s)$ denotes the number of times state $s$ has been visited, $N(s, z)$ denotes the number of times macro action$z$ has been chosen in state $s$, and $c$ is an exploration coefficient.

**Progressively Widening the State Space for Search.** Despite these powerful abstraction techniques, the search space remains challenging due to the underlying high-dimensional nature of the

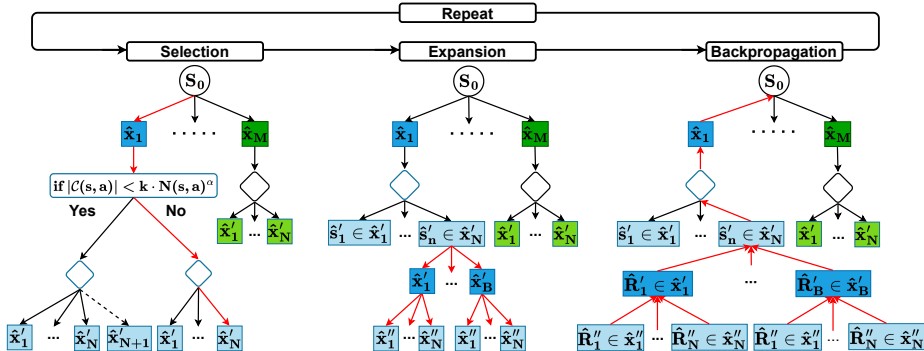

Figure 4: Illustration of our MCTS process for macro-level planning. The algorithm iteratively selects actions using the UCT policy, applies progressive widening to balance exploration and exploitation, performs parallel expansion of multiple macro actions and their potential outcomes, and backpropagates estimated Q-values to efficiently explore and refine the planning tree.

original state space, residual stochastic characteristics of transitions in the abstracted space, and the complexity of long-horizon planning scenarios. If we were to apply MCTS directly to this abstracted space, we would encounter two main issues: inefficient utilization of our pre-built search space, with the search potentially diverging prematurely into unexplored regions, and difficulty in building sufficiently deep trees for high-quality long-term decision-making, particularly in areas of high stochasticity or uncertainty (Couëtoux et al., 2011). Therefore, we use progressive widening to extend MCTS to incrementally expand the search tree. It balances the exploration of new states with the exploitation of already visited states based on two hyperparameters: $\alpha \in [0, 1]$ and $\epsilon \in \mathbb{R}^+$. Let $|\mathcal{C}(s, z)|$ denote the number of children for the state-action pair $(s, z)$. The key idea is to alternate between adding new child nodes and selecting among existing child nodes, depending on the number of times a state-action pair $(s, z)$ has been visited. A new state is added to the tree if $|\mathcal{C}(s, z)| < \epsilon \cdot N(s, z)^\alpha$, where $N(s, z)$ is the number of times the state-action pair has been visited. The hyperparameter $\alpha$ controls the propensity to select among existing children, with $\alpha = 0$ leading to always selecting among existing child and $\alpha = 1$ leading to vanilla MCTS behavior (always adding a new child). In this way, we could enhance our approach by efficiently utilizing the pre-built search space, prioritizing the exploration of promising macro actions while allowing for incremental expansion of the search tree. This technique enables our method to make quick decisions in an anytime manner, leveraging the cached information, and further refine the planning tree if additional time is available.

**Expansion.** In our approach, the *expansion* phase differs from standard MCTS by performing *parallel expansion* of multiple nodes from a leaf node. From the leaf node, a set of $B$ latent codes $\{z^{(i)}\}_{i=1}^B$ is sampled, each representing a distinct macro action, drawn from a latent transition model $p(z \mid s)$ to ensure diverse action space coverage. For each sampled macro action $z^{(i)}$, $N$ subsequent latent codes $\{z'^{(i,j)}\}_{j=1}^N$ are sampled according to $z'^{(i,j)} \sim p(z' \mid z^{(i)}, s)$, for $j = \{1, \ldots, N\}$, modeling potential outcomes and capturing the stochastic nature of macro actions. These latent transitions are then decoded to obtain the resulting next states $\{s'^{(i,j)}\}_{j=1}^N$ for each macro action. Finally, the search tree is expanded by adding all $L$ child nodes $\{(s'^{(i,j)}, z'^{(i,j)})\}_{j=1}^N$ for each macro action $z^{(i)}$ to the current leaf node $s$. This breadth-wise expansion enables simultaneous exploration of multiple promising macro actions, enhancing the diversity and comprehensiveness of the search and facilitating efficient exploration in complex environments.

**Backpropagation.** Following the expansion phase, where multiple macro actions are expanded simultaneously, the *backpropagation* step updates the estimated Q-values based on the return-to-go as shown in Fig.4.

## 4 EXPERIMENTS

The empirical evaluation of L-MAP consists of three sets of tasks from D4RL (Fu et al., 2020): gym locomotion control, AntMaze, and Adroit. We compare L-MAP to a range of prior offline RL

algorithms, including both model-free actor-critic methods (Kumar et al., 2020; Kostrikov et al., 2022) and model-based approaches (Rigter et al., 2023; Jiang et al., 2023; Janner et al., 2021). Our work is conceptually most related to the Trajectory Transformer (TT; Janner et al. (2021)) and the Trajectory Autoencoding Planner (TAP; Jiang et al. (2023)), which are model-based planning methods that predict and plan in continuous state and action spaces. These two baselines serve as our main points of comparison for deterministic environments.

To demonstrate L-MAP's ability to make performant decisions in stochastic environments, we compare it with One Risk to Rule Them All (1R2R; Rigter et al. (2023)), a risk-averse model-based algorithm designed for stochastic domains, and model-free actor-critic methods Conservative Q-Learning (CQL; Kumar et al. (2020)) and Implicit Q-Learning (IQL; Kostrikov et al. (2022)). We evaluate L-MAP on Stochastic MuJoCo tasks (Rigter et al., 2023), which serve as a proof of concept in the stochastic continuous control domain.

We then test L-MAP on Adroit, which presents a challenge with its high state and action dimensionality. Finally, we evaluate L-MAP on AntMaze, a sparse-reward continuous-control problem. In this task, L-MAP achieves similar performance to TT, surpassing model-free methods. Through these diverse evaluations, we aim to demonstrate L-MAP's versatility and effectiveness across different types of control problems, including stochastic environments, high-dimensional spaces, and sparse-reward scenarios. Additionally, we conduct an ablation study to analyze the impact of key components in L-MAP; detailed results of this study can be found in Appendix A.

**Hyperparameters** As for the L-MAP-specific hyperparameters, we set our macro action length to 3. The planning horizon in the raw action space is set to 9 for gym locomotion tasks and 15 for Adroit tasks. These horizons are either smaller or equal to those used in TT and TAP. Our choice of parameters is to ensure a control rate of approximately 10 Hz for locomotion tasks. For each task, we conduct experiments with 3 different training seeds, and each seed is evaluated for 20 episodes.

**Stochastic Mujoco** On the Stochastic MuJoCo tasks, with results presented in Table 1, L-MAP **consistently outperforms the model-based baselines, TAP and TT, across all datasets and environments**, demonstrating its superior capacity to handle stochasticity in continuous control tasks. Notably, L-MAP achieves the highest performance in multiple datasets for both the Hopper and Walker2D environments. When compared to 1R2R, a risk-averse model-based algorithm specifically designed for stochastic domains, L-MAP shows competitive or superior results in most cases. An exception is the Medium-Replay-High Hopper dataset, where 1R2R attains a higher score. This suggests that while L-MAP exhibits robustness across a variety of stochastic settings, there are specific scenarios where risk-averse strategies like 1R2R may hold an advantage. Additionally, L-MAP generally outperforms the model-free methods, CQL and IQL. However, CQL surpasses L-MAP in the Medium-Expert-Mod Hopper dataset. It is worth noting that L-MAP is the only method among all baselines that achieves performance comparable to CQL in this specific setting.

Table 1: Results for Stochastic MuJoCo.

| Dataset Type | Env | Model-Based | | | | Model-Free | |
| --- | --- | --- | --- | --- | --- | --- | --- |
| | | L-MAP | TAP | TT | 1R2R | CQL | IQL |
| Medium-Expert-Mod | Hopper | $106.11 \pm 2.16$ | $40.86 \pm 5.42$ | $56.10 \pm 3.33$ | $52.19 \pm 8.37$ | $\mathbf{106.17 \pm 2.16}$ | $60.61 \pm 3.46$ |
| Medium-Expert-Mod | Walker2D | $\mathbf{93.43 \pm 1.41}$ | $91.40 \pm 1.42$ | $80.93 \pm 2.60$ | $56.48 \pm 7.51$ | $91.44 \pm 1.44$ | $86.66 \pm 1.84$ |
| Medium-Mod | Hopper | $55.07 \pm 3.06$ | $43.64 \pm 2.25$ | $44.49 \pm 2.47$ | $\mathbf{65.24 \pm 3.31}$ | $49.92 \pm 3.00$ | $56.00 \pm 3.60$ |
| Medium-Mod | Walker2D | $52.94 \pm 1.57$ | $44.46 \pm 1.82$ | $43.61 \pm 2.15$ | $\mathbf{65.16 \pm 2.84}$ | $49.38 \pm 2.02$ | $48.82 \pm 2.31$ |
| Medium-Replay-Mod | Hopper | $\mathbf{52.30 \pm 2.65}$ | $38.10 \pm 3.22$ | $37.85 \pm 1.19$ | $22.82 \pm 2.08$ | $40.53 \pm 1.52$ | $49.12 \pm 3.38$ |
| Medium-Replay-Mod | Walker2D | $51.44 \pm 1.65$ | $43.49 \pm 2.27$ | $27.43 \pm 3.33$ | $\mathbf{52.23 \pm 2.22}$ | $40.24 \pm 1.67$ | $40.77 \pm 2.72$ |
| Medium-Expert-High | Hopper | $66.93 \pm 3.46$ | $37.31 \pm 3.66$ | $58.04 \pm 3.60$ | $37.99 \pm 2.71$ | $\mathbf{68.03 \pm 3.94}$ | $44.83 \pm 2.58$ |
| Medium-Expert-High | Walker2D | $\mathbf{97.18 \pm 2.08}$ | $91.09 \pm 2.78$ | $50.01 \pm 3.51$ | $32.38 \pm 4.55$ | $83.18 \pm 3.70$ | $68.61 \pm 3.33$ |
| Medium-High | Hopper | $\mathbf{55.32 \pm 3.56}$ | $43.93 \pm 2.66$ | $41.26 \pm 5.53$ | $33.99 \pm 0.92$ | $45.21 \pm 2.97$ | $49.69 \pm 2.47$ |
| Medium-High | Walker2D | $\mathbf{68.87 \pm 2.21}$ | $52.20 \pm 2.76$ | $59.84 \pm 5.03$ | $32.13 \pm 4.51$ | $61.49 \pm 3.24$ | $47.53 \pm 3.05$ |
| Medium-Replay-High | Hopper | $58.05 \pm 3.36$ | $48.69 \pm 2.97$ | $39.24 \pm 2.16$ | $\mathbf{68.25 \pm 3.78}$ | $51.70 \pm 3.09$ | $43.27 \pm 2.78$ |
| Medium-Replay-High | Walker2D | $\mathbf{65.87 \pm 3.07}$ | $55.15 \pm 3.29$ | $16.55 \pm 2.17$ | $65.63 \pm 3.41$ | $50.33 \pm 3.88$ | $45.13 \pm 2.38$ |
| **Mean** | | **68.63** | 52.53 | 46.28 | 48.71 | 61.47 | 53.42 |

**D4RL MuJoCo** On the deterministic MuJoCo tasks, particularly when compared to established model-free approaches such as CQL and IQL, L-MAP demonstrates **notable performance in environments like Walker2D and Hopper**, matching or exceeding these baselines even in dense reward scenarios as shown in Table 2. This highlights L-MAP's effectiveness across various task structures. When compared to TT, L-MAP consistently delivers comparable results. However, L-

MAP offers a significant practical advantage: its use of **temporal abstraction enables lower latency decision-making for equivalent planning horizons**, resulting in improved efficiency during deployment. Furthermore, L-MAP generally outperforms TAP, suggesting that even in deterministic environments, the expectation-based planning approach proves advantageous by accounting for stochasticity in the behavior policy. This leads to more robust policies and, consequently, superior results.

Table 2: Normalised results for D4RL MuJoCo-v2 following the protocol of Fu et al. (2020)

| Dataset Type | Env | Model-Based | | | | Model-Free | |
|---|---|---|---|---|---|---|---|
| | | **L-MAP** | **TAP** | **TT** | **1R2R** | **CQL** | **IQL** |
| Medium-Expert | HalfCheetah | $92.14 \pm 0.26$ | $86.40 \pm 2.22$ | $\mathbf{95.0 \pm 0.2}$ | $93.99 \pm 1.40$ | 91.6 | 86.7 |
| Medium-Expert | Hopper | $105.74 \pm 2.24$ | $85.55 \pm 3.83$ | $\mathbf{110.0 \pm 2.7}$ | $57.40 \pm 6.06$ | 105.4 | 91.5 |
| Medium-Expert | Walker2D | $109.35 \pm 0.08$ | $105.32 \pm 2.03$ | $101.9 \pm 6.8$ | $73.18 \pm 6.29$ | 108.8 | **109.6** |
| Medium | HalfCheetah | $45.50 \pm 0.10$ | $44.73 \pm 0.39$ | $46.9 \pm 0.4$ | $\mathbf{73.45 \pm 0.15}$ | 44.4 | 47.4 |
| Medium | Hopper | $\mathbf{73.90 \pm 1.91}$ | $69.14 \pm 2.33$ | $61.1 \pm 3.6$ | $55.49 \pm 3.99$ | 58.0 | 66.3 |
| Medium | Walker2D | $\mathbf{80.31 \pm 1.20}$ | $51.75 \pm 3.30$ | $79.0 \pm 2.8$ | $55.69 \pm 4.97$ | 72.5 | 78.3 |
| Medium-Replay | HalfCheetah | $38.45 \pm 0.80$ | $40.83 \pm 0.72$ | $41.9 \pm 2.5$ | $\mathbf{63.85 \pm 0.19}$ | 45.5 | 44.2 |
| Medium-Replay | Hopper | $91.18 \pm 0.56$ | $80.92 \pm 3.79$ | $91.5 \pm 3.6$ | $89.67 \pm 1.92$ | **95.0** | 94.7 |
| Medium-Replay | Walker2D | $81.04 \pm 2.62$ | $72.32 \pm 3.26$ | $82.6 \pm 6.9$ | $\mathbf{90.67 \pm 1.98}$ | 77.2 | 77.2 |
| **Mean** | | **79.73** | 70.77 | 78.88 | 72.60 | 77.60 | 77.32 |

**Adroit Control** In the Adroit robotic control tasks, which are characterized by their high-dimensional state and action spaces, our proposed method, L-MAP, **demonstrates strong and competitive performance** as shown in Table 3. Across the Human, Cloned, and Expert datasets, L-MAP exhibits notable effectiveness compared to both model-based approaches (TAP and TT) and model-free methods (CQL, IQL, and Behavior Cloning (BC)[1]).

Table 3: Adroit robotic hand control results.

| Dataset Type | Env | Model-Based Approaches | | | Model-Free Approaches | | |
|---|---|---|---|---|---|---|---|
| | | **L-MAP** | **TAP** | **TT** | **CQL** | **IQL** | **BC** |
| Human | Pen | $\mathbf{76.26 \pm 8.58}$ | $66.86 \pm 8.41$ | 36.4 | 37.5 | 71.5 | 34.4 |
| Human | Hammer | $1.71 \pm 0.12$ | $1.57 \pm 0.09$ | 0.8 | **4.4** | 1.4 | 1.5 |
| Human | Door | $\mathbf{11.24 \pm 1.11}$ | $9.51 \pm 1.10$ | 0.1 | 9.9 | 4.3 | 0.5 |
| Human | Relocate | $0.09 \pm 0.02$ | $0.06 \pm 0.01$ | 0.0 | **0.2** | 0.1 | 0.0 |
| Cloned | Pen | $\mathbf{60.68 \pm 7.88}$ | $46.44 \pm 7.54$ | 11.4 | 39.2 | 37.3 | 56.9 |
| Cloned | Hammer | $\mathbf{2.43 \pm 0.29}$ | $1.32 \pm 0.12$ | 0.5 | 2.1 | 2.1 | 0.8 |
| Cloned | Door | $13.22 \pm 1.34$ | $\mathbf{13.45 \pm 1.43}$ | $-0.1$ | 0.4 | 1.6 | $-0.1$ |
| Cloned | Relocate | $\mathbf{0.15 \pm 0.13}$ | $-0.23 \pm 0.01$ | $-0.1$ | $-0.1$ | $-0.2$ | $-0.1$ |
| Expert | Pen | $\mathbf{126.60 \pm 5.60}$ | $112.16 \pm 6.57$ | 72.0 | 107.0 | – | 85.1 |
| Expert | Hammer | $127.16 \pm 0.29$ | $\mathbf{128.79 \pm 0.52}$ | 15.5 | 86.7 | – | 125.6 |
| Expert | Door | $105.24 \pm 0.10$ | $\mathbf{105.86 \pm 0.08}$ | 94.1 | 101.5 | – | 34.9 |
| Expert | Relocate | $\mathbf{107.57 \pm 0.76}$ | $106.21 \pm 1.61$ | 10.3 | 95.0 | – | 101.3 |
| **Mean (All)** | | **51.40** | 49.33 | 20.08 | 40.32 | 14.76 | 36.73 |
| **Mean (Non-Expert)** | | **18.79** | 17.37 | 6.13 | 11.70 | 14.76 | 11.74 |

In the Human dataset, which includes suboptimal human demonstrations, L-MAP achieves the highest score in the Door environment and performs well in other tasks. Although IQL leads in the Pen task and CQL leads in the Hammer and Relocate tasks, L-MAP maintains competitive results, particularly surpassing TT and BC in most environments. This suggests that L-MAP effectively utilizes suboptimal data to make robust decisions in complex settings. For the Cloned dataset, which contains a mix of optimal and suboptimal trajectories, L-MAP secures top performance in the Pen and Relocate tasks. In the Expert dataset, comprised of optimal demonstrations, L-MAP attains the highest scores in the Pen and Relocate environments while remaining competitive in the Hammer and Door tasks. Overall, L-MAP achieves the highest average score of 51.40 across all datasets and environments, and 18.79 across non-expert datasets, highlighting its effectiveness in handling varying levels of data optimality. Furthermore, the experimental results indicate that L-MAP effectively

---

[1]We included Behavior Cloning (BC) as an additional baseline since the original 1R2R method was not evaluated for Adroit tasks.

manages the complexities of high-dimensional Adroit environments. Incorporating more action information into the single token does not detract from performance; instead, it appears to enhance the model's ability to learn nuanced temporal dependencies required for successful task execution.

**AntMaze** In the AntMaze environments—a set of sparse-reward continuous-control tasks where an agent must navigate a robotic ant to a target location, L-MAP demonstrates strong and competitive performance as shown in Table 4. These tasks are particularly challenging due to the sparse rewards and the presence of suboptimal trajectories that lead to various goals other than the target position used during testing.

Table 4: Performance comparison on AntMaze environments. This evaluation demonstrates that our approach can achieve comparable performance to TT with a separate Q network, while being more efficient during sampling and decision-making.

| Dataset Environment | BC | CQL | IQL | TT (+Q) | TAP | L-MAP |
|---|---|---|---|---|---|---|
| Umaze AntMaze | 54.6 | 74.0 | 87.5 | **100.0 ± 0.0** | 78.33 ± 5.32 | 93.33 ± 3.22 |
| Medium-Play AntMaze | 0.0 | 61.2 | 71.2 | **93.3 ± 6.4** | 43.33 ± 6.40 | 75.00 ± 6.85 |
| Medium-Diverse AntMaze | 0.0 | 53.7 | 70.0 | **100.0 ± 0.0** | 30.00 ± 5.92 | 88.33 ± 4.14 |
| Large-Play AntMaze | 0.0 | 15.8 | 39.6 | 66.7 ± 12.2 | 63.33 ± 6.22 | **78.33 ± 5.32** |
| Large-Diverse AntMaze | 0.0 | 14.9 | 47.5 | 60.0 ± 12.7 | 66.67 ± 6.09 | **81.67 ± 5.00** |
| **Mean** | 10.92 | 43.92 | 55.16 | **84.00** | 56.33 | 83.33 |

Similar to TAP, our approach integrates goal positions into the observation space, allowing it to condition trajectory generation on specific goals. This conditioning narrows the focus of sampled trajectories towards the target direction, simplifying the planning process. Instead of using the IQL critic for value estimation, L-MAP leverages Monte Carlo planning to provide refined value estimates. This alternative approach avoids the additional computational cost of sampling with a separate Q-network, as required by TT (+Q).

Our method achieves an average success rate of 83.33% across all AntMaze environments, which is comparable to the 84.00% average of TT (+Q). Notably, L-MAP outperforms TT (+Q) in the more complex Large-Play and Large-Diverse environments, achieving success rates of 78.33% and 81.67% respectively, compared to TT (+Q)'s 66.7% and 60.0%. This indicates that L-MAP is particularly effective in larger mazes where navigation complexity is higher. While TT (+Q) attains perfect success rates in smaller environments like Umaze and Medium-Diverse, L-MAP still performs exceptionally well with success rates of 93.33% and 88.33% in these settings. This consistency suggests that our method is robust across different scales of environment complexity.

## 5 RELATED WORK

Recent advancements in reinforcement learning focus on learning temporally extended action primitives to reduce decision-making horizons and improve learning efficiency. Both model-free and model-based methods leverage temporal abstraction to manage task complexity.

Model-free methods such as CompILE (Kipf et al., 2019), RPL (Gupta et al., 2019), OPAL (Ajay et al., 2021), ACT (Zhao et al., 2023), and PRISE (Zheng et al., 2024) leverage temporal abstraction in various ways. For instance, CompILE learns latent codes representing variable-length behavior segments, enabling cross-task generalization. RPL employs a hierarchical policy architecture to simplify long-horizon tasks by decomposing them into sub-policies. OPAL introduces a continuous space of primitive actions to reduce distributional shift in offline RL, enhancing policy robustness. PRISE applies sequence compression to learn variable-length action primitives, improving behavior cloning by capturing essential behavioral patterns. These approaches demonstrate the versatility of temporal abstraction in addressing different challenges in reinforcement learning, particularly in managing the complexity inherent in sequential decision-making.

From a model-based perspective, recent work has treated reinforcement learning as a sequence modeling problem, utilizing Transformer architectures to model entire trajectories of states, actions, rewards, and values. This approach is exemplified by methods like Trajectory Transformer (TT) (Zhou et al., 2020), and TAP (Jiang et al., 2023). TAP, in particular, shares conceptual similarities with

our proposed method, L-MAP, in its use of efficient planning solutions for complex action spaces. These sequence modeling approaches have shown promise in capturing long-term dependencies and handling the variability in trajectories, but they often face challenges in stochastic environments where the outcome is not solely determined by the agent's actions. As highlighted by Paster et al. (2022), reinforcement learning via supervised learning methods may replicate suboptimal actions that accidentally led to good outcomes due to environmental randomness. To address this issue, they proposed ESPER, a solution inspired by the decision transformer framework (Chen et al., 2021). ESPER mitigates the influence of stochasticity on policy learning in discrete action spaces by clustering trajectories and conditioning on average cluster returns.

From a theoretical perspective, several foundational works have studied continuous-space RL via Hamilton-Jacobi-Bellman equations. For example, Kim et al. (2021) grounded Q-learning and DQN in this theory, characterizing optimal control without explicit optimization, Munos (2000) established convergence results using viscosity solutions, and Han et al. (2017) employed deep learning to solve high-dimensional PDEs via backward stochastic differential equations. While providing crucial theoretical foundations, these works focused on deterministic environments or required perfect knowledge about the dynamics of the environment.

Our approach also has interesting connections to robust RL, though with key distinctions. While robust MDPs (Iyengar, 2005; Nilim & Ghaoui, 2005) deal with varying transition kernels chosen adversarially from uncertainty sets, our work focuses on learning and planning with a fixed transition kernel in an offline setting where environmental stochasticity is captured through learned models. Early robust RL addressed planning with known dynamics in tabular settings (Xu & Mannor, 2010), and generalizing to continuous, high-dimensional spaces is challenging (Lim & Autef, 2019). Our temporal abstraction could complement robust RL by providing structured transition functions, potentially integrating classical robust RL planning into high-dimensional environments.

From a planning perspective, our work relates to methods like MuZero (Schrittwieser et al., 2020), stochastic MuZero (Antonoglou et al., 2022), and Vector Quantized Models for Planning (Ozair et al., 2021), which primarily operate in discrete action spaces and online settings, limiting their applicability to continuous control tasks in offline RL. MuZero Unplugged (Schrittwieser et al., 2021) extended MuZero to the offline setting and adapted to low-dimensional continuous action spaces using factorized policy representations (Tang & Agrawal, 2020). However, scaling to high-dimensional action spaces is challenging due to computational infeasibility and imprecise action selection (Luo et al., 2023b). Additionally, MuZero Unplugged focuses on deterministic environments and may struggle in highly stochastic continuous settings.

Our method, L-MAP, extends these concepts to high-dimensional continuous action spaces by effectively handling stochasticity and complexity. Using an encoder to group similar state-macro-action pairs and reconstructing return-to-go estimates via a decoder within the VQ-VAE framework, L-MAP captures essential dynamics while abstracting unnecessary details. This approach models future returns more accurately in stochastic settings. Combined with planning algorithms, L-MAP refines expected return estimates, bridging the gap between temporal abstraction techniques and robust performance in stochastic environments. Our latent code representation and transition model reduce the need to learn separate policy, dynamics, and value components in the offline setting, increasing planning efficiency and accounting for environmental stochasticity, thereby enhancing generalization across complex tasks.

## 6 DISCUSSION AND LIMITATIONS

In conclusion, we introduced the Latent Macro Action Planner (L-MAP), which leverages temporal abstractions learned with a state-conditioned VQ-VAE to construct a discrete latent space of macro-actions. This approach enables efficient planning in high-dimensional continuous action spaces within stochastic environments. Future directions include exploring transfer learning to handle new tasks, and adapting L-MAP to online learning scenarios through strategies such as risk-averse exploration (Luo et al., 2024). These enhancements would enable continuous improvement and help tackle more complex challenges, ultimately improving generalization and efficiency in complex, real-world settings.

## 7 ACKNOWLEDGEMENTS

This material is based upon work sponsored by the National Science Foundation (NSF) under Grant CNS-2238815 and by the Defense Advanced Research Projects Agency (DARPA) under the *Assured Neuro Symbolic Learning and Reasoning* program. Results presented in this paper were obtained using the Chameleon testbed supported by the National Science Foundation. Any opinions, findings, conclusions, or recommendations expressed in this material are those of the authors and do not necessarily reflect the views of the NSF or the DARPA.

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

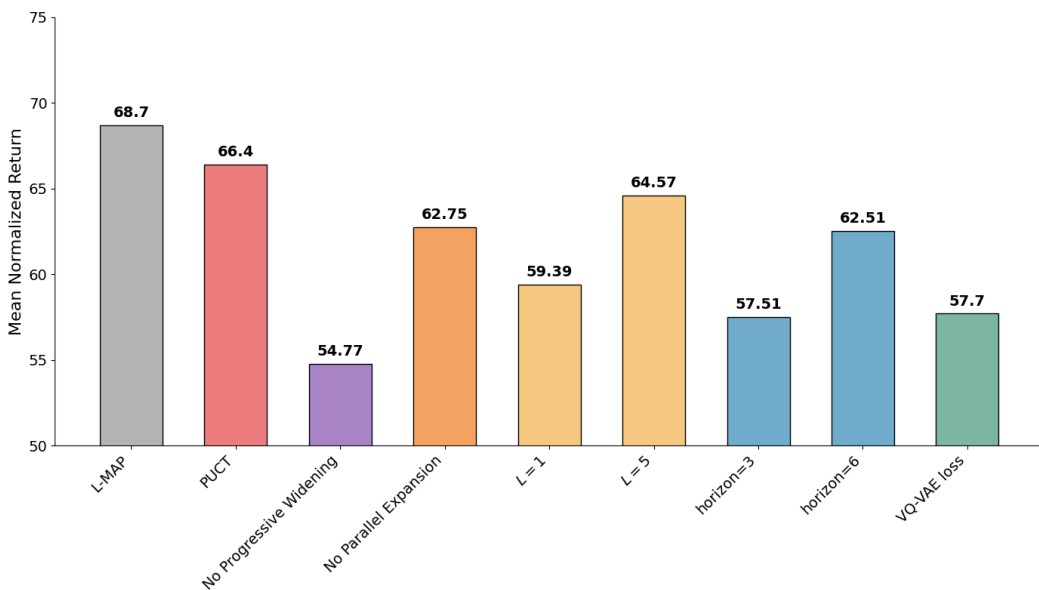

Figure 5: Results of ablation studies, where the height of the bar is the mean normalized scores on high noise gym locomotion control tasks.

## A    ABLATION STUDY

We present analyses and ablations of key hyperparameters such as macro action length, planning horizon, the use of pUCT (Silver et al., 2017) versus UCT, and the effect of our customized VQ-VAE loss function. Figure 5 summarizes the results from ablation studies conducted on high-noise stochastic MuJoCo tasks.

**Macro Action Length**

We tested macro action lengths $L = 1$, $L = 3$, and $L = 5$ to evaluate their impact on L-MAP's performance. The highest mean score of 68.7 was achieved with $L = 3$. Increasing $L$ to 5 reduced the mean score to 64.57, while decreasing it to 1 further dropped it to 59.39. This indicates that a macro action length of 3 optimally balances temporal abstraction and adaptability. A moderate length allows the model to capture important action sequences while remaining responsive to environmental changes. Shorter lengths may fail to model temporal dependencies effectively, while longer lengths may hinder quick adaptation in stochastic environments.

**Planning Horizon**

We assessed the effect of planning horizon by varying the number of planning steps in L-MAP. Reducing the planning horizon to 3 steps (expanding a single latent variable) decreased the mean score to 57.51, compared to 68.7 with the default longer planning horizon. This demonstrates that a longer planning horizon significantly enhances performance by enabling the model to better anticipate future events and handle uncertainty in high-noise stochastic environments.

**Tree Search Algorithm: UCT vs. pUCT**

We compared standard UCT and pUCT as tree search algorithms in L-MAP. UCT achieved a mean score of 68.7, slightly outperforming pUCT, which scored 66.4. While both methods are effective, UCT performs marginally better in this context. A possible explanation is that pUCT leverages a learned prior policy to guide exploration, making it sensitive to the quality of the prior. If the prior policy is suboptimal, pUCT may be less effective due to this dependency.

**VQ-VAE Loss Function**  We compared our loss function with the standard loss function without masking (mean scores: 68.7 vs 57.7). Our approach outperforms the standard loss by focusing primarily on state and action during vector quantization. This results in less skewed reconstructed

returns and a more coherent latent space, accurately capturing action and state distributions. Consequently, the model generates more reliable latent representations for reconstruction.

**Progressive Widening**

We evaluated the impact of progressive widening on MAP's performance. Removing progressive widening led to a significant drop in the mean score from 68.70 to 54.77. This substantial decrease demonstrates the importance of controlled state space expansion during planning for a large search space. Progressive widening enables MAP to balance between exploiting existing states in the pre-built search space and incrementally adding new states. Without progressive widening, the search suffers from excessive branching, making it difficult to build sufficiently deep trees for meaningful planning in areas of high stochasticity.

**Parallel Expansion**

We assessed the contribution of parallel expansion by comparing L-MAP's performance with and without this feature. Removing parallel expansion reduced the mean score from 68.7 to 62.75, yielding performance similar to reducing the planning horizon to six steps. This comparison reveals that parallel expansion primarily affects the algorithm's ability to efficiently explore the search space. Given the same number of MCTS iterations, removing parallel expansion results in less exploration of possible trajectories, reducing the algorithm's planning capability to that of a shorter horizon. This demonstrates that parallel expansion is crucial for maximizing the effectiveness of each MCTS iteration by enabling broader simultaneous exploration of potential outcomes.

## B  ADDITIONAL STOCHASTIC ENVIRONMENT EXPERIMENTS: HIV TREATMENT AND CURRENCY EXCHANGE

Table 5: Results for HIV Treatment and Currency Exchange.

| Env | Model-Based Approaches | | | | Model-Free Approaches | |
|-----|---------|------|------|------|------|------|
|  | **L-MAP** | **TAP** | **TT** | **1R2R** | **CQL** | **IQL** |
| HIV | $59.08 \pm 1.96$ | $54.95 \pm 1.98$ | $54.46 \pm 3.30$ | $56.45 \pm 2.17$ | $\mathbf{59.74 \pm 1.11}$ | $34.1 \pm 1.2$ |
| Currency | $\mathbf{106.78 \pm 5.00}$ | $89.72 \pm 3.90$ | $79.28 \pm 2.61$ | $78.52 \pm 2.08$ | $93.96 \pm 1.69$ | $89.41 \pm 2.83$ |

The HIV Treatment environment, originally introduced by Ernst et al. (2006), simulates treatment planning where an agent controls two drug types (RTI and PI) in a 6-dimensional state space representing cell and virus concentrations. The stochasticity arises from varying drug efficacy at each step. The Currency Exchange environment, based on the Optimal Liquidation problem (Almgren & Chriss, 2001; Bao & Liu, 2019), involves converting currency under stochastic exchange rates that follow an Ornstein-Uhlenbeck process. Both environments were adapted by Rigter et al. (2023) to the offline RL setting, with datasets collected using partially trained and random policies respectively.

For the HIV Treatment domain, L-MAP and CQL achieve comparable strong performance (59.08 $\pm$ 1.96 and 59.74 $\pm$ 1.11 respectively), outperforming other baselines. In the Currency Exchange environment, L-MAP substantially outperforms all other approaches, achieving a score of 106.78 $\pm$ 5.00 compared to the next best performer CQL at 93.96 $\pm$ 1.69. This superior performance demonstrates L-MAP's versatility across different types of stochastic environments.

## C  LATENT SPACE ANALYSIS

To empirically demonstrate the uncertainties introduced by non-injective mappings, behavior policy, and environmental stochasticity, we generate heatmaps representing the transition probabilities between latent codes. We focus on the Hopper environment and consider three datasets: `medium-expert`, `medium`, and `medium-replay`, in both deterministic and stochastic settings. The heatmaps are constructed by encoding the state-macro-action pairs into latent codes using our learned representation and visualizing the transition probabilities between these codes.

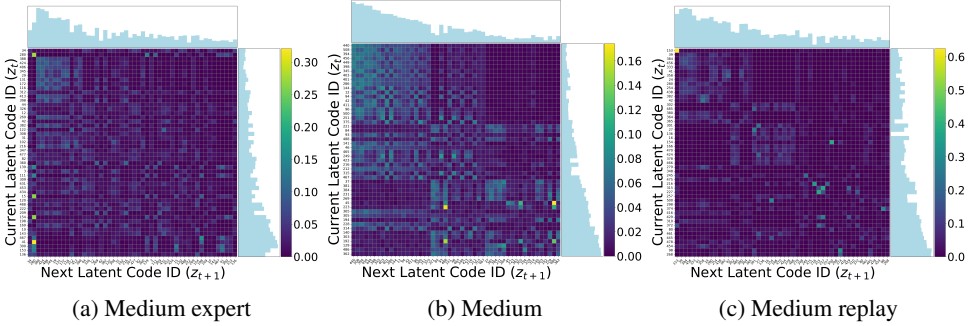

(a) Medium expert        (b) Medium        (c) Medium replay

Figure 6: Heatmaps for Deterministic Hopper Environment (Top 50 Frequent Latent Codes). In each heatmap, the intensity of the color at position $(i, j)$ represents the probability of transitioning from the current latent code $z_t = i$ to the next latent code $z_{t+1} = j$. The accompanying histograms display the frequency of each latent code occurring across the dataset with the learned encoder as the current ($z_t$, right histogram) and next ($z_{t+1}$, top histogram) codes. The observed spread in the heatmaps indicates that, despite the deterministic nature of the environment, transitions from a single $z_t$ lead to multiple $z_{t+1}$.

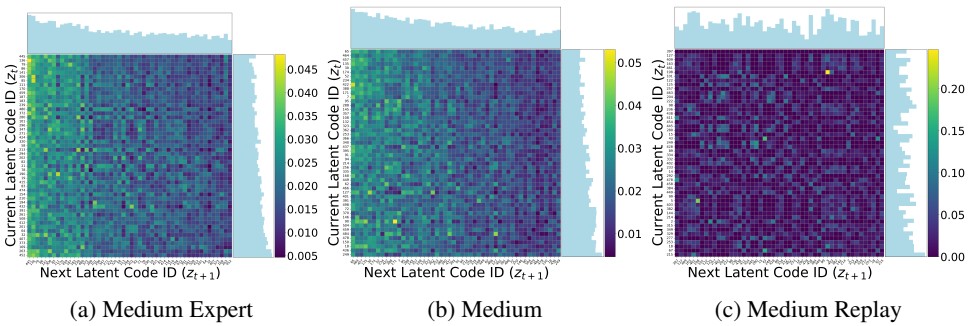

(a) Medium Expert        (b) Medium        (c) Medium Replay

Figure 7: Heatmaps for Stochastic Hopper Environment (Top 50 Frequent Latent Codes). The observed spread in the heatmaps indicates that inherent environmental stochasticity further contributes to transitions from a single $z_t$ leading to multiple $z_{t+1}$.

## C.1 DETERMINISTIC ENVIRONMENT HEATMAPS

In analyzing the heatmaps for deterministic environments as shown in Fig. 6, it becomes evident that transitions from a current latent code $z_t$ to multiple next latent codes $z_{t+1}$ are not strictly deterministic. This observed spread in transitions originates from two primary sources: the **non-injective nature of the learned representation** and the **stochasticity of the behavior policy** employed during data collection.

First, the **non-injective mapping** of the encoder function $f_{enc}$ may result in multiple distinct high-dimensional state-macro-action pairs being mapped to the same latent code as shown in the histograms of Fig.6. Specifically, for different state-macro-action pairs $x_t^{(1)} = (s_t^{(1)}, m_t^{(1)})$ and $x_t^{(2)} = (s_t^{(2)}, m_t^{(2)})$, it is possible that:

$$f_{enc}(x_t^{(1)}) = f_{enc}(x_t^{(2)}) = z_t,$$

even though $x_t^{(1)} \neq x_t^{(2)}$. Consequently, their corresponding next state-macro-action pairs $x_{t+1}^{(1)}$ and $x_{t+1}^{(2)}$ may differ, potentially leading to different next latent codes upon encoding:

$$z_{t+1}^{(1)} = f_{enc}(x_{t+1}^{(1)}), \quad z_{t+1}^{(2)} = f_{enc}(x_{t+1}^{(2)}), \quad \text{with} \quad z_{t+1}^{(1)} \neq z_{t+1}^{(2)}.$$

Second, because the **behavior policy** $\pi_b$ used for data collection may be stochastic, it introduces variability in the selection of macro-actions at both the current and subsequent time steps. Given a state $s_t$, the behavior policy determines the macro-action $m_t$ as follows:

$$m_t \sim \pi_b(m \mid s_t).$$

This stochastic selection can result in different macro-actions $m_t^{(1)}$ and $m_t^{(2)}$ being chosen from the same state $s_t$, which naturally introduces stochasticity. Note that even if the encoder maps both $x_t^{(1)} = (s_t, m_t^{(1)})$ and $x_t^{(2)} = (s_t, m_t^{(2)})$ to the same latent code $z_t$:

$$f_{\text{enc}}(x_t^{(1)}) = f_{\text{enc}}(x_t^{(2)}) = z_t.$$

the next states $s_{t+1}^{(1)}$ and $s_{t+1}^{(2)}$ might differ, even though the environment dynamics $T_{\text{env}}$ are deterministic, i.e.,

$$s_{t+1}^{(1)} = T_{\text{env}}(s_t, m_t^{(1)}), \quad s_{t+1}^{(2)} = T_{\text{env}}(s_t, m_t^{(2)}), \quad \text{with} \quad s_{t+1}^{(1)} \neq s_{t+1}^{(2)}.$$

These different next states lead to different next state-macro-action pairs:

$$x_{t+1}^{(1)} = (s_{t+1}^{(1)}, m_{t+1}^{(1)}), \quad x_{t+1}^{(2)} = (s_{t+1}^{(2)}, m_{t+1}^{(2)}).$$

Upon encoding, they may yield different next latent codes:

$$z_{t+1}^{(1)} = f_{\text{enc}}(x_{t+1}^{(1)}), \quad z_{t+1}^{(2)} = f_{\text{enc}}(x_{t+1}^{(2)}), \quad \text{with} \quad z_{t+1}^{(1)} \neq z_{t+1}^{(2)}.$$

Therefore, even in a deterministic environment, the combination of a non-injective encoder and a stochastic behavior policy introduces variability in the latent transitions. The heatmaps for deterministic environments empirically demonstrate this spread, showing that each $z_t$ does not map deterministically to a single $z_{t+1}$ but rather to a distribution of possible next latent codes.

### C.2 STOCHASTIC ENVIRONMENT HEATMAPS

The heatmaps for stochastic environments as shown in Fig. 7 exhibit a more pronounced spread in transition probabilities. This inherent environmental stochasticity means that for a given $s_t$ and $m_t$, there are multiple possible next states $s_{t+1}$, leading to a wider distribution of next latent codes $z_{t+1}$ upon encoding. When combined with the non-injective mapping of the encoder and the stochasticity of the behavior policy, the uncertainties in the latent transitions are further amplified.

### C.3 THE IMPACT OF $L_1$ REGULARIZATION ON REPRESENTATION FIDELITY

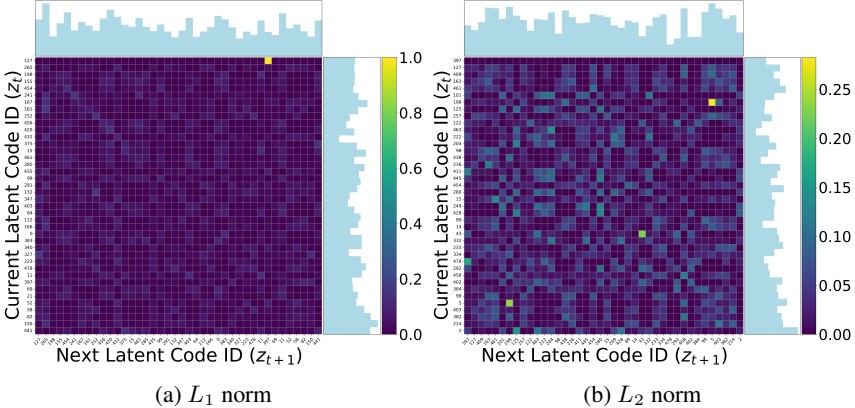

(a) $L_1$ norm

(b) $L_2$ norm

Figure 8: Transition Probability Heatmaps for Medium-Replay Datasets from the Stochastic Hopper Environment (Top 50 Frequent Latent Codes). Left: Heatmap depicting transition probabilities when embeddings are regularized using the L1 norm. Right: Heatmap illustrating transition probabilities under L2 norm regularization.

The heatmaps shown in Fig.8 reveal distinct patterns between transition probabilities for latent codes encoded by encoders trained with $L_1$ and $L_2$ norm regularization in the latent space. The $L_2$ norm demonstrates more distributed transition probabilities, with multiple moderate-probability transitions (shown as light blue dots) for each current state, indicating the encoder preserves more granular information. In contrast, the $L_1$ norm exhibits highly deterministic transitions for certain latent codes, shown by the predominantly dark purple background with the bright yellow spot approaching probability 1.0. This suggests that the encoder trained with $L_1$ regularization tends to collapse dissimilar inputs into the same latent code, leading to less nuanced representations.

# D ANALYSIS OF PERFORMANCE TRENDS WITH INCREASING STOCHASTICITY

Table 6: Hopper Environment Results with Increasing Stochasticity

| Dataset Type | Env | Model-Based | | | | Model-Free | |
|---|---|---|---|---|---|---|---|
| | | L-MAP | TAP | TT | 1R2R | CQL | IQL |
| **Deterministic** | | | | | | | |
| Medium-Expert | Hopper | $105.74 \pm 2.24$ | $85.55 \pm 3.83$ | **$110.0 \pm 2.7$** | $57.40 \pm 6.06$ | 105.4 | 91.5 |
| Medium | Hopper | **$73.90 \pm 1.91$** | $69.14 \pm 2.33$ | $61.1 \pm 3.6$ | $55.49 \pm 3.99$ | 58.0 | 66.3 |
| Medium-Replay | Hopper | $91.18 \pm 0.56$ | $80.92 \pm 3.79$ | $91.5 \pm 3.6$ | $89.67 \pm 1.92$ | 95.0 | 94.7 |
| **Mean (Deterministic)** | | **90.27** | 78.54 | 87.53 | 67.52 | 86.13 | 84.17 |
| **Moderate Stochasticity** | | | | | | | |
| Medium-Expert-Mod | Hopper | $106.11 \pm 2.16$ | $40.86 \pm 5.42$ | $56.10 \pm 3.33$ | $52.19 \pm 8.37$ | **$106.17 \pm 2.16$** | $60.61 \pm 3.46$ |
| Medium-Mod | Hopper | $55.07 \pm 3.06$ | $43.64 \pm 2.25$ | $44.49 \pm 2.47$ | **$65.24 \pm 3.31$** | $49.92 \pm 3.00$ | $56.00 \pm 3.60$ |
| Medium-Replay-Mod | Hopper | **$52.30 \pm 2.65$** | $38.10 \pm 3.22$ | $37.85 \pm 1.19$ | $22.82 \pm 2.08$ | $40.53 \pm 1.52$ | $49.12 \pm 3.38$ |
| **Mean (Moderate Stochasticity)** | | **71.16** | 40.87 | 46.15 | 46.75 | 65.54 | 55.24 |
| **High Stochasticity** | | | | | | | |
| Medium-Expert-High | Hopper | $66.93 \pm 3.46$ | $37.31 \pm 3.66$ | $58.04 \pm 3.60$ | $37.99 \pm 2.71$ | **$68.03 \pm 3.94$** | $44.83 \pm 2.58$ |
| Medium-High | Hopper | **$55.32 \pm 3.56$** | $43.93 \pm 2.66$ | $41.26 \pm 5.53$ | $33.99 \pm 0.92$ | $45.21 \pm 2.97$ | $49.69 \pm 2.47$ |
| Medium-Replay-High | Hopper | $58.05 \pm 3.36$ | $48.69 \pm 2.97$ | $39.24 \pm 2.16$ | **$68.25 \pm 3.78$** | $51.70 \pm 3.09$ | $43.27 \pm 2.78$ |
| **Mean (High Stochasticity)** | | **60.10** | 43.31 | 46.18 | 46.74 | 54.98 | 45.93 |

This section examines how L-MAP and baseline methods respond to increasing levels of stochasticity in the Hopper environment. Table 6 presents the performance metrics across deterministic, moderate, and high stochasticity settings.

In the **deterministic** setting, L-MAP achieves a mean score of **90.27**, indicating strong performance and outperforming all other model-based methods. Among the baselines, TT attains a mean of 87.53, TAP achieves 78.54, and 1R2R scores 67.52. The model-free methods CQL and IQL also perform well, with mean scores of 86.13 and 84.17, respectively. The high scores across all methods suggest that the deterministic environment poses minimal challenges, allowing both L-MAP and the baselines to excel.

As the environment introduces **moderate stochasticity**, L-MAP's mean performance decreases to **71.16**, reflecting a reduction of approximately $21\%$ from its deterministic performance. The model-based baselines experience larger declines; TAP's mean drops to 40.87 (a $48\%$ reduction), TT's to 46.15 (a $47\%$ reduction), and 1R2R's to 46.75 (a $31\%$ reduction). The model-free methods also suffer performance losses; CQL's mean decreases to 65.54 (a $24\%$ reduction), and IQL's to 55.24 (a $34\%$ reduction). Despite the reductions, L-MAP maintains a higher mean score than all baselines in this setting, indicating better resilience to moderate stochasticity among both model-based and model-free methods.

In the setting of **high stochasticity**, L-MAP's mean further decreases to **60.10**, representing a total reduction of about $33\%$ from the deterministic case. The model-based baselines continue to show declining trends; TAP's mean falls to 43.31 (a $45\%$ reduction), TT's to 46.18 (a $47\%$ reduction), and 1R2R's to 46.74 (a $31\%$ reduction). The model-free methods also see further decreases; CQL's mean drops to 54.98 (a $36\%$ reduction), and IQL's to 45.93 (a $45\%$ reduction). While all methods experience performance degradation, L-MAP consistently outperforms the model-based baselines TAP and TT, and maintains an edge over the model-free methods CQL and IQL. The performance of L-MAP shows relatively better robustness among the baselines.

The overall trend indicates that increasing stochasticity adversely affects all methods, but L-MAP's performance diminishes at a slower rate compared to the other model-based methods. These results suggest that L-MAP is more robust to stochastic variations in the environment than most of the baseline methods, particularly the model-based ones.

# E PLANNING HYPERPARAMETERS

For all environments, we utilize the following hyperparameters for sampling during the search process: $\alpha = 0.1$ and $\epsilon = 1$, which determine the exploration rate of progressive widening; and set

the number of Monte Carlo Tree Search (MCTS) iterations to 100. Detailed parameters for each environment are presented in Table 7.

Table 7: Planning Hyperparameters

| Environment | M | N | B | $\lambda$ | $\gamma$ |
|---|---|---|---|---|---|
| Stochastic MuJoCo | 16 | 4 | 4 | 0.5 | 0.99 |
| D4RL MuJoCo | 16 | 4 | 4 | 0.5 | 0.99 |
| Adroit | 10 | 2 | 4 | 0.5 | 0.99 |
| AntMaze | 16 | 2 | 4 | 0.5 | 0.998 |
| Currency | 32 | 4 | 4 | 0.5 | 0.99 |
| HIV Treatment | 5 | 4 | 4 | 1.0 | 0.99 |

