# OpenReview forum: "Scalable Decision-Making in Stochastic Environments through Learned Temporal Abstraction"
_ICLR.cc/2025/Conference — ICLR 2025 Spotlight_

### Official Review · Reviewer_Wb4D · 2024-11-04

**Soundness:** 3
**Presentation:** 3
**Contribution:** 3
**Rating:** 8
**Confidence:** 4

**Summary:**

Paper focusses on stochastic planning in offline RL settings (where the data was collected by a stochastic behavioral policy). To avoid the stochasticitiy related with continuous action spaces, paper focuses on a temporally extended (latent) macro-action policy that is obtained via a vector-quantized variational autoencoder (VQ-VAE).

**Strengths:**

- I love the integration of primitive actions in the raw state space which is then abstracted temporally via encoders (VQ-VAE) to generate a discrete latent code. Sticking a transformer on the latent codes sounds good for creating a distribution over state and primitive action representations.

+ I think it's nice that the authors were thoughtful enough to add a conditionally widening MCTS tree to the search in the latent space to further aid the autoregressive and stochastic continuous L-MAP policy.

- Perhaps supply the examples of stochastic environments that motivated the question you asked in the Introduction.

**Weaknesses:**

+ I love the motivation in the introduction that many environments appear stochastic but a couple of examples where offline RL has been tried and where significant gaps in efficiency has been noticed would further strengthen the authors' argument.

+ While it's true that continuous action spaces and stochasticity in environments may make planning horizons longer, the authors would do well to mention the role of robust control or robust RL in mitigating some of these challenges whilst highlighting how their proposed method complements these existing methods;
 - In this sentiment, I am interested to see a motivation for why the authors decided to embark on the temporal abstractions proposal;

 + There is a significant body of work in continuous space RL in the realm of the viscosity solutions to Hamilton-Jacobi equations. The authors should discuss some of these in their intro/related work. Example texts:

  - A Study of Reinforcement Learning in the Continuous Case by the Means of viscosity Solutions: https://www.ri.cmu.edu/pub_files/pub1/munos_remi_1999_3/munos_remi_1999_3.pdf

  - Solving high-dimensional partial differential equations using deep learning: https://www.pnas.org/doi/pdf/10.1073/pnas.1718942115

+ Please define what sg signifies in equation (2)!

**Questions:**

+ How is the length of the macro action chosen? Predetermined or randomly assigned in an on-the-go fashion?

+ What are examples of real-world problems with high-dimensional continuous action spaces in the opinion of the author(s)?

+ I think you missed a couple of connecting words between lines 76 and 77. Do you want to fix those?

+ I am curious to see how this offline L-MAP policy behaves in off-policy RL settings. Can you provide an example or two?

+ L127-128: "particularly for online approaches"; why was this qualification needed seeing we are dealing only with offline RL in this paper?

+ How did you determine the instanteneous rewards, r_i, for the computation of R in equation (1)?

+ Lines 167-171, what loss function did the authors use in the reconstruction process? There have been instances in reconstruction like this when the choice of loss makes a huge difference in the reconstruction. See https://arxiv.org/pdf/2311.03534

+ What happens if instead of using the l2 norm on the embeddings regularizer, you try the l1 norm? Does it improve the repreesentation's fidelity?

+ Can you supply a proof, empirical results, ot citation for the claim on lines 213-217?

---

> ### Author Response · Authors · 2024-11-21
> **Response 1 for Reviewer Wb4D**
>
> We sincerely thank the reviewer for the valuable feedback. We have addressed each of the comments point-by-point below and are available to engage in further discussions should any additional questions arise.
> >1. Perhap supply the examples of stochastic environments that motivated the question you asked in the Introduction.
> I love the motivation in the introduction that many environments appear stochastic but a couple of examples where offline RL has been tried and where significant gaps in efficiency has been noticed would further strengthen the authors' argument.
>
> We have added more concrete motivations into the introduction to strengthen our argument.
>
>  >2. While it's true that continuous action spaces and stochasticity in environments may make planning horizons longer, the authors would do well to mention the role of robust control or robust RL in mitigating some of these challenges whilst highlighting how their proposed method complements these existing methods;
> In this sentiment, I am interested to see a motivation for why the authors decided to embark on the temporal abstractions proposal;
>
>
> Thank you for highlighting the connection to robust RL. Let us clarify the key distinctions and potential connections:
> First, Robust MDPs as defined in seminal works [1][2] deal with non-fixed transition kernels that can be adversarially chosen from an uncertainty set at each timestep. In contrast, our work focuses on learning and planning with a fixed transition kernel in an offline setting where environmental stochasticity is captured through our learned models.
> However, we believe L-MAP could significantly complement robust RL approaches. Early robust RL work primarily tackled planning problems with known transition probability dynamics in tabular settings [3]. Generalizing these approaches to continuous, high-dimensional spaces has remained challenging [4]. Our method could potentially address this gap by providing structured transition functions through learned temporal abstractions. This offers a potential direction for integrating classical robust RL planning approaches into our framework for robust decision-making in high-dimensional environments.
> From an offline RL perspective, robust RL aims to learn policies that remain effective despite model parameter mismatches between training and testing environments [5]. The integration of robust RL principles with our temporal abstraction approach could potentially yield methods that are both computationally efficient and robust to model misspecification. We believe this represents an exciting direction for future research. We have included a discussion on robust RL and its relation to our work in the related work section of our paper. Thanks!
>
> [1] Garud Iyengar. Robust Dynamic Programming. Math. Oper. Res., 30:257-280, 2005.
>
> [2] Arnab Nilim and Laurent El Ghaoui. Robust Control of Markov Decision Processes with Uncertain Transition Matrices. Oper. Res., 53:780-798, 2005.
>
> [3] Huan Xu and Shie Mannor. Distributionally Robust Markov Decision Processes. Math. Oper. Res., 37:288-300, 2010.
>
> [4] Shiau Hong Lim and Arnaud Autef. Kernel-Based Reinforcement Learning in Robust Markov Decision Processes. Proceedings of the 36th International Conference on Machine Learning (ICML), PMLR 97:3973-3981, 2019.
>
> [5] Kishan Panaganti, Zaiyan Xu, Dileep M. Kalathil, and Mohammad Ghavamzadeh. Robust Reinforcement Learning using Offline Data. arXiv preprint arXiv:2208.05129, 2022.
>
> >3. There is a significant body of work in continuous space RL in the realm of the viscosity solutions to Hamilton-Jacobi equations. The authors should discuss some of these in their intro/related work. Example texts:
> A Study of Reinforcement Learning in the Continuous Case by the Means of viscosity Solutions: https://www.ri.cmu.edu/pub_files/pub1/munos_remi_1999_3/munos_remi_1999_3.pdf
> Solving high-dimensional partial differential equations using deep learning: https://www.pnas.org/doi/pdf/10.1073/pnas.1718942115
>
> Thank you for the suggestion! We have included a discussion of these works in the related work section, focusing on continuous-space reinforcement learning framed through Hamilton-Jacobi equations.
>
> >4. Please define what sg signifies in equation (2)!
>
> We have added a definition of it in the paper which denotes the stopgradient operator.
>
> >5. How is the length of the macro action chosen? Predetermined or randomly assigned in an on-the-go fashion?
>
> The length of the macro action is determined before training.

---

> ### Author Response · Authors · 2024-11-21
> **Response 2 for Reviewer Wb4D**
>
> >6. What are examples of real-world problems with high-dimensional continuous action spaces in the opinion of the author(s)?
>
> Thanks for the insightful question! Dexterous manipulation with high degrees of freedom (DOF) robotic hands like the Shadow Dexterous Hand (26 DOF) [1] represents a key real-world example of high-dimensional continuous action spaces. This is particularly important for embodied AI, as dexterous manipulation is essential for agents to physically interact with their environment in human-like ways.
>
> Besides dexterous manipulation, electric vehicle charging control [2] is another emerging real-world problem with high-dimensional continuous action spaces. For a charging station with N EVSEs (charging ports), the action space is N-dimensional (e.g, ranging 16 to 96) where each dimension controls one EVSE's charging power to optimize energy trading while minimizing battery degradation and transformer stress.
>
> [1] Liu, Y., Chen, W., Bai, Y., Li, G., Gao, W., & Lin, L. (2024). Aligning Cyber Space with Physical World: A Comprehensive Survey on Embodied AI. CoRR, abs/2407.06886.
>
> [2] Truong, V. B., & Le, L. B. (2024). Electric vehicle charging design: The factored action based reinforcement learning approach. Applied Energy, 359, 122737.
>
> >7. I think you missed a couple of connecting words between lines 76 and 77. Do you want to fix those?
>
> Thanks for pointing this out, I have fixed it.
>
> >8. I am curious to see how this offline L-MAP policy behaves in off-policy RL settings. Can you provide an example or two?
>
> Thank you for your insightful question. Offline RL can be viewed as a subset of off-policy RL since both learn from data generated by a behavior policy [3], but offline RL prohibits additional data collection and online policy improvement. While L-MAP was primarily designed for offline RL, exploring its behavior in standard off-policy RL settings where the agent can interact with the environment is an interesting direction. In such settings, L-MAP can be adapted to leverage both the fixed dataset and new experiences collected online.
>
> For example, one can leverage Monte Carlo Tree Search (MCTS) for online data collection with some domain-specific constraints such as being risk-averse for exploration and potentially discovering strategies not present in the initial dataset. Incorporating streaming data for model updates allows the agent to continuously refine its latent model with new observations, enhancing adaptability and leading to improved policy performance over time. Extending L-MAP in this way could enhance its applicability and performance in dynamic environments where online data collection is possible.
>
> [3] Levine, S., Kumar, A., Tucker, G., & Fu, J. (2020). Offline Reinforcement Learning: Tutorial, Review, and Perspectives on Open Problems. CoRR, abs/2005.01643.
>
> >9. L127-128: "particularly for online approaches"; why was this qualification needed seeing we are dealing only with offline RL in this paper?
>
> Although our models are trained offline, the planning algorithm operates online during deployment.
>
> >10. How did you determine the instanteneous rewards, r_i, for the computation of R in equation (1)?
>
> Thank you for your question. In the offline reinforcement learning setup, the instantaneous rewards $r_i$ are directly obtained from the provided trajectories. For a given time step $t$, the cumulative discounted reward  $R_t$ is computed as: $R_t = \sum_{i=t}^{T} \gamma^{i - t} r_i$. Therefore $R_t$ directly includes the instantaneous reward $r_t$ at the current time step $t$.
>
> >11.  what loss function did the authors use in the reconstruction process?
>
> We used the mean squared error (MSE) as the loss function in our reconstruction process.
>
> >12. What happens if instead of using the l2 norm on the embeddings regularizer, you try the l1 norm? Does it improve the representation's fidelity?
>
> We have investigated this and found that $L_1$ norm actually leads to less desirable representations. **Our analysis shows that $L_1$ regularization tends to create more deterministic latent coded transitions, collapsing dissimilar inputs into the same latent codes.** In contrast, $L_2$ norm better preserves the granularity of representations by maintaining more distributed transition probabilities. **We have included detailed transition probability heatmaps in the paper's appendix C.3 to support these findings.**

---

> ### Author Response · Authors · 2024-11-21
> **Response 3 for Reviewer Wb4D**
>
> >13. Can you supply a proof, empirical results, or citation for the claim on lines 213-217?
>
> Thank you for this question.We have conducted empirical analysis to demonstrate why transitions in the latent space are not deterministic, even in deterministic settings. Our analysis reveals three key sources of stochasticity: (1) the non-injective nature of the learned representation, where multiple distinct state-macro-action pairs can be mapped to the same latent code; (2) the stochasticity of the behavior policy used during data collection; and (3) environmental stochasticity when present. **We have added detailed transition probability heatmaps and empirical analysis in appendix C.1 and C.2 to support these claims. The heatmaps clearly show that each latent code transitions to multiple following latent codes with varying probabilities, providing empirical evidence for our claim.**

---

> ### Author Response · Authors · 2024-11-25
> **Follow-up**
>
> Dear reviewer Wb4D
>
> We sincerely thank you for your thorough review and constructive feedback, which has significantly contributed to improving our paper. Your detailed insights have been invaluable in helping us refine our work. We hope that our responses have sufficiently addressed all the issues you highlighted in your review. Please do not hesitate to let us know if there is anything further we could do to improve your confidence in our work and address any remaining concerns.
>
> Best regards,
>
> Authors of paper #13144

---

> ### Comment · Reviewer_Wb4D · 2024-11-27
> **Re: Response to Questions and Weaknesses**
>
> Dear Author(s),
>
> + Thank you for correcting the prose in the paper where I pointed out a need for correction.
>
> + I love that the new write-up includes a literature review discussion about on robust RL and HJB equations.
>
> + This proposal of yours, which is unique because it addresses stochasticity of dynamical systems, seems interesting. However, I am not at all convinced that the joint space of the shadow robot  hand is stochastic. What introduces this stochasticity in your opinion? The comment about electric charge distribution  for EVs seems interesting and I actually glanced through the cited paper you referenced to see where the stochasticity arises in the decision-making process about how much charge to give to a vehicle whilst balancing it with the power supply. I struggle to see the stochastic connection. Did you mean that the stochasticity is in the underlying dynamics or in the resulting policy? What is the source of the stochasticity?
>
> + But more importantly, what is the nature of the stochasticity that your algorithm addresses? Is it in the dynamics or the control? Is the noise Gaussian or Brownian or some other stochasticity arising from other colored noise? I think you need to make a strong case for the definitiveness of your contribution, its relation to the types of systems in which it may be used (motivation)
>
> + I appreciate the answers to my other clarifying questions. I find it interesting that the $L_1$ loss function actually leads to a degradation in performance. Well done!

---

> > ### Comment · Reviewer_Wb4D · 2024-11-27
> >
> > I have updated the score I gave for the contributions of the paper. I am willing to revise my overall score if my questions from above are answered.

---

> > ### Author Response · Authors · 2024-11-28
> > **Response 1 to the new questions**
> >
> > Thank you very much for your invaluable feedback and thorough review! Please find our answers to the new questions below:
> >
> > > This proposal of yours, which is unique because it addresses stochasticity of dynamical systems, seems interesting. However, I am not at all convinced that the joint space of the shadow robot hand is stochastic. What introduces this stochasticity in your opinion? The comment about electric charge distribution for EVs seems interesting and I actually glanced through the cited paper you referenced to see where the stochasticity arises in the decision-making process about how much charge to give to a vehicle whilst balancing it with the power supply. I struggle to see the stochastic connection. Did you mean that the stochasticity is in the underlying dynamics or in the resulting policy? What is the source of the stochasticity?
> >
> > Thank you for the interesting questions! Prior work has examined how measurement uncertainty can cause sensor observations to deviate from true states, as demonstrated in robotic systems like the Segway platform [1]. To counter this uncertainty from sensor noise variability, there is prior work on incorporating stochastic measurement uncertainty into control formulations to achieve robustness [2]. While the Shadow Hand's joint-space mechanics are fundamentally deterministic, potential measurement noise in the sensing system can introduce stochasticity.
> >
> > The stochasticity in the EV charging problem lies in the environment dynamics. Prior work [3] explicitly models this environmental stochasticity through three key random processes following truncated Gaussian distributions (Table 3 in the paper [3]): EV arrival times, departure times, and initial battery states. While the charging actions specify exact power allocations, the effectiveness of these actions depends on these uncertain factors.
> >
> > Due to these uncertainties, it's hard to come up with a charging schedule that maximizes profits because the outcome of any charging decision directly depends on random future events. For example, if the station decides to charge an EV slowly to save electricity costs, the EV might depart earlier than expected, leaving it with insufficient charge and leading to lost revenue and customer dissatisfaction. Conversely, if the station decides to charge an EV rapidly, the EV might end up staying much longer than expected, meaning expensive electricity was used unnecessarily when cheaper rates were available later.
> >
> > [1]Cosner, R. K., Singletary, A. W., Taylor, A. J., Molnár, T. G., Bouman, K. L., & Ames, A. D. (2021). Measurement-robust control barrier functions: Certainty in safety with uncertainty in state. IEEE/RSJ International Conference on Intelligent Robots and Systems, IROS 2021
> >
> > [2]Yang, S., Pappas, G. J., Mangharam, R., & Lindemann, L. (2023). Safe perception-based control under stochastic sensor uncertainty using conformal prediction. 62nd IEEE Conference on Decision and Control, CDC 2023
> >
> > [3] Truong, V. B., & Le, L. B. (2024). Electric vehicle charging design: The factored action based reinforcement learning approach. Applied Energy

---

> > > ### Comment · Reviewer_Wb4D · 2024-11-29
> > >
> > > Thank you for the clarification!

---

> > ### Author Response · Authors · 2024-11-28
> > **Response 2 to the new questions**
> >
> > > But more importantly, what is the nature of the stochasticity that your algorithm addresses? Is it in the dynamics or the control? Is the noise Gaussian or Brownian or some other stochasticity arising from other colored noise? I think you need to make a strong case for the definitiveness of your contribution, its relation to the types of systems in which it may be used (motivation)
> >
> > **Our algorithm is primarily designed to address stochastic dynamics**, although we also show that we can handle stochasticity introduced by the non-injective nature of learned representations even in deterministic settings (please see the updated experimental analysis on the latent space in Fig.6 in the updated draft). With respect to stochastic dynamics, **we have demonstrated the effectiveness of our algorithm across several domains, incorporating various types of stochastic dynamics (e.g., uniform noise, Gaussian variability, and Ornstein-Uhlenbeck processes)**, following the settings introduced in the paper [1]:
> >
> > **Stochastic MuJoCo Environments**: Random perturbation forces are applied following a random walk model to simulate external disturbances like wind.
> >
> > **HIV Treatment Domain (appendix section 5)**: Variability in patient responses is modeled by allowing drug efficacy to vary stochastically with a 15% standard deviation from the nominal value.
> >
> > **Currency Exchange Domain  (appendix section 5)**: Exchange rates evolve according to an Ornstein-Uhlenbeck stochastic process, modeling the volatility observed in financial markets.
> >
> > The effectiveness of our approach in these domains demonstrates that our algorithm can handle various forms of uncertainty. Our method achieves this through its data-driven approach., i.e., as long as the training datasets capture the underlying stochastic dynamics, no specific tailoring to noise models is required. This is crucial for real-world applications such as robots, drones, and autonomous vehicles where online data collection is expensive and potentially unsafe [2]. By learning from existing data, our method enables effective model-based planning while handling various forms of uncertainty.
> >
> > [1] Rigter, M., Lacerda, B., & Hawes, N. (2023). One Risk to Rule Them All: A Risk-Sensitive Perspective on Model-Based Offline Reinforcement Learning. In Advances in Neural Information Processing Systems 36 (NeurIPS 2023).
> >
> > [2] Levine, S., Kumar, A., Tucker, G., & Fu, J. (2020). Offline Reinforcement Learning: Tutorial, Review, and Perspectives on Open Problems. CoRR, abs/2005.01643.
> >
> > **Please do not hesitate to let us know if our answers fully or partially address your questions. We look forward to further discussion with you. Thank you!**

---

> > > ### Comment · Reviewer_Wb4D · 2024-11-29
> > >
> > > Thank you for the clarifications!

---

### Official Review · Reviewer_UcYT · 2024-11-06

**Soundness:** 3
**Presentation:** 4
**Contribution:** 4
**Rating:** 8
**Confidence:** 2

**Summary:**

This paper proposes Latent Macro Action Planner (L-MAP), an offline RL algorithm designed for high-dimensional continuous action spaces in stochastic environments. L-MAP learns a discrete latent space of temporally extended macro-actions to reduce the action dimensionality and simplify the planning process. A causal transformer is learnt in tandem as a latent transition model, predicting future latent codes. Planning is performed using MCTS with progressive widening. Experiments show that L-MAP outperforms model based and model free baselines in stochastic environments while being on par in deterministic environments, highlighting its versatility.

**Strengths:**

- Comprehensive experiments: The paper provides a suite of experiments covering multiple tasks with good model based and model free baselines.
- The inclusion of the Adroit hand task showcases the scalability of the proposed method to high dimensional problems.
- The use of macro-actions and a pre-constructed search space allows for faster decision-making.
- Additionally, L-MAP provides a way to trade off latency and longer planning depending on the user requirements.
- The ablation study provides insights into the importance of different hyperparameters and design choices, strengthening the paper's conclusions.

**Weaknesses:**

See questions below.

**Questions:**

- Could you explain the source of stochasticity in the Mujoco tasks used in the paper? Is it primarily from the dataset generating policy or are the environments themselves modified in some way?
- Why is L-MAP less performant in the deterministic environments? Is there an underlying theoretical reason for lower performance in deterministic settings given that it was more consistently the best performing algorithm in the stochastic setting?
- Could the authors study how L-MAP compares to the baselines with increasing levels of stochasticity(in either the dataset or transition dynamics) ? This can be for a single environment/task.

---

> ### Author Response · Authors · 2024-11-21
> **Response 1 for Reviewer UcYT**
>
> We deeply appreciate the reviewer's insightful feedback and are thankful for the recognition of our technical contributions and analyses. Please find our detailed responses to the comments below. We are available to engage in further discussions should any additional questions arise.
> > Could you explain the source of stochasticity in the Mujoco tasks used in the paper? Is it primarily from the dataset generating policy or are the environments themselves modified in some way?
>
> The sources of stochasticity in our study can be categorized into two major categories.
>
> 1) Note that environments such as D4RL MuJoCo Benchmarks (HalfCheetah, Hopper, Walker2D) [1] are deterministic by design. However, any scalable approach for high-dimensional state and action spaces invariably performs some dimensionality reduction, which also introduces stochasticity through the representation learning process, even in deterministic settings. This stochasticity primarily emerges from the non-injective nature of learned representations, mainly through encoding techniques such as VQ-VAE, where multiple state-macro-action pairs can map to the same latent code. This naturally leads to probabilistic transitions in latent space. Additionally, the behavior policies used for data collection introduce variability. This variability is further captured by non-injective mapping.
>
> 2) The modified stochastic MuJoCo benchmarks [2] are stochastic by design. The environmental stochasticity is introduced through perturbation during transitions. We add different noise levels (moderate noise, high noise) to simulate real-world uncertainties like environmental disturbances or actuator noise. As a result, the robot's dynamics become inherently probabilistic due to these modifications.
>
> **We have updated the paper with empirical analysis and transition probability heatmaps (Appendix C.1 and C.2). The results demonstrate that even in deterministic environments, a single latent code can transition to multiple subsequent latent codes with varying probabilities.**
>
> [1] Fu, J., Kumar, A., Nachum, O., Tucker, G., & Levine, S. (2020). D4RL: Datasets for Deep Data-Driven Reinforcement Learning. CoRR, abs/2004.07219. Fu, J., Kumar, A., Nachum, O., Tucker, G., & Levine, S. (2020).
>
> [2] Rigter, M., Lacerda, B., & Hawes, N. (2023). One Risk to Rule Them All: A Risk-Sensitive Perspective on Model-Based Offline Reinforcement Learning. In Advances in Neural Information Processing Systems 36 (NeurIPS 2023).
>
> > Why is L-MAP less performant in the deterministic environments? Is there an underlying theoretical reason for lower performance in deterministic settings given that it was more consistently the best performing algorithm in the stochastic setting?
>
> Thank you for the insightful question! While L-MAP does not consistently outperform other baselines across all data types and environments in deterministic D4RL MuJoCo-v2, it achieves the highest mean performance (79.73) across all environments and data types, demonstrating its overall robustness. **The performance variation between different environments in a deterministic setting can be attributed to the datasets exhibiting varying degrees of stochasticity due to the behavior policies used in data collection.** We have added an analysis on the latent space in Appendix C revealing that certain environments and datasets, such as the “Hopper-v2 with medium” dataset (Fig.6(b)), display more stochasticity between current and subsequent latent codes. In these cases, L-MAP's MCTS-based planning approach becomes particularly effective, leading to the best performance compared to other baselines. **The benefits of using MCTS in deterministic environments thus appear to be closely tied to the stochasticity present in the dataset itself rather than just the environmental dynamics.**

---

> ### Author Response · Authors · 2024-11-21
> **Response 2 for Reviewer UcYT**
>
> > Could the authors study how L-MAP compares to the baselines with increasing levels of stochasticity(in either the dataset or transition dynamics) ? This can be for a single environment/task.
>
> Building on our previous discussion about dataset stochasticity, our analysis provides insights into how L-MAP performs under different levels of stochasticity. Taking the Hopper environment as an example, our transition heatmap analysis in Appendix C shows that the medium dataset (Fig.6(b)) exhibits more probabilistic transitions between latent codes compared to other datasets. In this setting, where the underlying data distribution shows higher stochasticity despite the deterministic environment, L-MAP achieves the best performance compared to other baselines.
>
> For stochastic MuJoCo environments, we observe similar patterns. In the highly stochastic Hopper environment, our analysis shows that medium-expert (Fig.7 (a)) and medium datasets (Fig.7 (b)) demonstrate significantly more probabilistic transitions between latent codes compared to medium-replay datasets (Fig.7 (c)). This increased stochasticity in the latent space transitions aligns with L-MAP's performance improvements over other model-based baselines in these settings.
>
> **We also added an analysis of performance trends with increasing stochasticity in the Hopper environment in Section D of appendix and included the corresponding data in Table 6 of the paper.** The analysis demonstrates that L-MAP exhibits greater robustness compared to baseline methods. While all methods experience declines in performance as stochasticity intensifies, L-MAP's performance diminishes at a slower rate. Specifically, from the deterministic setting to moderate stochasticity, L-MAP's mean performance decreases by approximately 21%, whereas model-based baselines like TAP and TT suffer larger reductions of 48% and 47%, respectively. Model-free methods such as CQL and IQL also experience reductions of 24% and 34%. In the high stochasticity setting, L-MAP's total performance reduction from the deterministic case is about 33%, which is less than the declines observed for TAP (45%), TT (47%), CQL (36%), and IQL (45%). These findings indicate that L-MAP maintains higher relative performance under increasing stochastic variations.

---

> ### Author Response · Authors · 2024-11-25
> **Follow-up**
>
> Dear reviewer UcYT
>
> We sincerely thank you for your thorough review, valuable feedback, and insightful questions! Your detailed insights have been invaluable in helping us refine our work. We hope that our responses have sufficiently addressed all the questions you highlighted in your review. Please do not hesitate to let us know if there is anything further we could do to improve your confidence in our work and address any remaining concerns.
>
> Best regards,
>
> Authors of paper #13144

---

> > ### Comment · Reviewer_UcYT · 2024-11-26
> >
> > The additional experiments by the authors are very helpful in addressing my questions. I do not have any remaining concerns.

---

> ### Author Response · Authors · 2024-11-26
> **Thank you!**
>
> Thank you for your positive feedback! We're pleased that our rebuttal addressed your concerns. It would be greatly appreciated if you could consider revisiting the confidence score when you have a chance. Thank you again for the thorough review and insightful feedback!

---

### Official Review · Reviewer_q9DP · 2024-11-08

**Soundness:** 3
**Presentation:** 2
**Contribution:** 3
**Rating:** 6
**Confidence:** 3

**Summary:**

The authors propose a model-based method for planning in stochastic environments called Latent Macro Action Planner (L-MAP). Their method first learns an encoder-decoder architecture to encode return-to-go estimates, states, and a sequence of actions, which they call macro actions. The learned discrete codes are then used to learn a state-conditioned latent prior $p(z|s)$ which is used to seed a modified Monte Carlo Tree Search (MCTS). The MCTS uses progressive widening to expand the search based on the prior initialization using a heuristic based on the number of children in the tree and the number of times a state-action node is visited and uses the prior to sample children for leaf nodes.

The authors evaluate their method in two stochastic offline reinforcement learning tasks, Hopper and Walker, and compare their method in the standard Hopper, Walker, HalfCheetah, AntMaze, and Adroit deterministic tasks. They also provide ablations over some of their hyperparameters used in their method.

**Strengths:**

The authors propose pre-constructing the latent search space using their learned prior and show benefits over searching without this pre-constructed prior, which is interesting.

**Weaknesses:**

The proposed algorithm has many layers of complexity and it is unclear which components contribute to the method's success. The two transformer-based baselines that they compare against do not utilize any temporal action compression, begging the question of if they would benefit similarly from predicting macro actions. While the authors ablate over macro action length and planning horizon, they do not provide any experiments supporting their decisions to utilize UCT for selection, progressive widening, or parallel expansion, and do not provide the values for hyperparameters $M,N,B,\alpha$ in their experiments.

Their results also leave a lot to be desired. The only environment where the method separates itself from existing methods is in the stochastic environments, but they only test in two such environments.

**Questions:**

How does your method compare to versions of TT or TAP that learn macro actions?
In Table 4, you include a column named "DT". Is this decision transformer and if so, why do you include it in the results when it is not mentioned in the paper?

---

> ### Author Response · Authors · 2024-11-21
> **Response 1 for Reviewer q9DP**
>
> We greatly value the reviewer’s constructive feedback and have provided detailed, point-by-point responses below. We are available to discuss any further questions or concerns you may have.
>
> > The proposed algorithm has many layers of complexity and it is unclear which components contribute to the method's success.
>
> We appreciate the reviewer's concern regarding the complexity of L-MAP and the opportunity to provide further clarification. Below, we outline the key concepts and their roles in decision-making within stochastic continuous environments. We have also added this discussion to the appendix.
>
> **Temporal Abstraction (Key Objective: efficiency and action space reduction)**: Macro-actions provide an essential abstraction for managing high-dimensional continuous action spaces to compress the planning horizon over extended action sequences as single units. The latent macro action space can be much smaller than the raw action space since it only captures plausible actions in the dataset, reducing decision complexity while maintaining adaptability.
>
> **Continuous-to-Discrete Mapping (Key Objective: dimensionality reduction)**: We achieve this goal through a State-Conditioned VQ-VAE. This core component discretizes the continuous state-macro-action space into a discrete latent representation, building on prior works' [1,2] recognition that learned state-conditioned discretization can maintain high granularity with relatively few discrete actions.
>
> **Latent Prior Model (Key Objective: Efficient sampling and transition modeling)**: The learned state-conditioned prior serves dual purposes - efficiently sampling plausible macro-actions and modeling transitions in the latent space, which is crucial for planning initialization.
>
> **Planning (Key Objective: heuristic sequential search in the latent space)**: We propose a modified MCTS algorithm that uses the learned prior to seed the initial search space and employs progressive widening to balance between exploitation of promising actions and exploration of alternatives, enabling effective decision optimization in the latent space.
>
> Each component addresses the specific challenge for decision-making within stochastic continuous environments, working together to enable efficient planning. We would greatly appreciate it if the reviewer seeks further discussion on any specific component, allowing us to provide more detailed explanations of their necessity and function. The paper already presents an ablation study that shows the use of temporal abstractions, and we will update the paper with another ablation study before the end of the rebuttal deadline.
>
> [1] Jiang, Z., Zhang, T., Janner, M., Li, Y., Rocktäschel, T., Grefenstette, E., & Tian, Y. (2023). Efficient Planning in a Compact Latent Action Space. ICLR 2023.
>
> [2] Luo, J., Dong, P., Wu, J., Kumar, A., Geng, X., & Levine, S. (2023). Action-Quantized Offline Reinforcement Learning for Robotic Skill Learning. CoRL 2023.
>
> >2. The two transformer-based baselines that they compare against do not utilize any temporal action compression, begging the question of if they would benefit similarly from predicting macro actions.
>
> Thank you for your comment, but we would like to respectfully point out a misunderstanding. **TAP does utilize temporal abstraction** and can be viewed as an improvement over TT, which has been found effective but inefficient.
>
> Our work (L-MAP) fundamentally differs from both TAP and TT in that we address the challenge of planning in stochastic environments, while both TAP and TT were primarily designed for and evaluated in deterministic settings. **In particular, we focus on handling stochasticity that arises from the environment and the representation learning process.**

---

> ### Author Response · Authors · 2024-11-21
> **Response 2 for Reviewer q9DP**
>
> > While the authors ablate over macro action length and planning horizon, they do not provide any experiments supporting their decisions to utilize UCT for selection, progressive widening, or parallel expansion, and do not provide the values for hyperparameters.
>
> We have now added **Section E to the appendix which provides a list of planning hyperparameters** used in our experiments, including values for M (initial samples), N (subsequent samples), B (latent codes per state), and $\alpha$ for each environment.
>
> **Regarding our choice of UCT for selection**: UCT [1] is the standard and commonly used bandit-based strategy for MCTS as it provides a theoretically grounded approach to balance exploration and exploitation through the Upper Confidence Bounds principle. While alternatives exist, such as P-UCT [2], which is included in the ablation study, UCT has been consistently shown to be effective across many domains and applications.
>
> Note that progressive widening is a widely used approach for scaling MCTS to large search spaces. **We use progressive widening as the search space is extremely vast and presents two major challenges**: (1) inefficient utilization of the pre-built space with premature divergence into unexplored regions, and (2) difficulty in building sufficiently deep trees for high-quality decision-making, as noted in prior work [3]. Progressive widening allows our method to dynamically balance rapid planning using learned abstractions with more exhaustive exploration when computational resources permit.
>
> We implement the **parallel expansion strategy primarily as an infrastructure for computational optimization** to speed up the exploration of the search space given a limited number of MCTS iterations[4].
>
> We are conducting more ablation studies and will add the results to the paper on progressive widening and parallel expansion before the end of the rebuttal period.
>
> [1] Kocsis, L., & Szepesvári, C. (2006). Bandit Based Monte-Carlo Planning. ECML 2006: Proceedings of the 17th European Conference on Machine Learning, LNCS, Vol. 4212, pp. 282–293. Springer.
>
> [2] Silver, D., Schrittwieser, J., Simonyan, K., Antonoglou, I., Huang, A., Guez, A., ... & Hassabis, D. (2018). A general reinforcement learning algorithm that masters chess, shogi, and Go through self-play. Science, 362(6419), 1140–1144.
>
> [3] Sokota, S., Ho, C., Ahmad, Z. F., & Kolter, J. Z. (2021). Monte Carlo Tree Search With Iteratively Refining State Abstractions. Advances in Neural Information Processing Systems 34 (NeurIPS 2021)
>
> [4] Chaslot, G., Winands, M. H. M., & van den Herik, H. J. (2008). Parallel Monte-Carlo Tree Search. Computers and Games (CG 2008), Lecture Notes in Computer Science, 5131, 60–71.
>
> > How does your method compare to versions of TT or TAP that learn macro actions?
>
> We point out that TAP, which builds on TT, **does learn temporal abstractions** through state, action, and reward aggregation across multiple time steps. However, one of our key contributions is how our method (L-MAP) handles temporal abstraction compared to TAP, the critical distinction being how these methods handle stochasticity. TAP's approach of aggregating states across time steps can obscure the underlying stochasticity in the environment's dynamics. In contrast, L-MAP is specifically designed to capture and work with stochasticity, both from the environment and the representation learning process. Our method maintains a clear representation of probabilistic transitions, allowing for more robust planning in both stochastic and deterministic environments.
>
> >In Table 4, you include a column named "DT". Is this decision transformer and if so, why do you include it in the results when it is not mentioned in the paper?
>
> Thank you for pointing this out. You are correct that "DT" in Table 4 refers to the Decision Transformer. Since DT did not perform as well as other baselines [5,6], we decided to exclude it from our discussion to focus on the most relevant comparisons. The inclusion of "DT" in Table 4 was an oversight, and we have now removed it for clarity. We appreciate your attention to this detail and have updated the paper accordingly.
>
> [5] Janner, M., Li, Q., & Levine, S. (2021). Offline reinforcement learning as one big sequence modeling problem. In Advances in Neural Information Processing Systems 34 (NeurIPS 2021)
>
> [6] Kostrikov, I., Nair, A., & Levine, S. (2022). Offline reinforcement learning with implicit Q-learning. In Proceedings of the Tenth International Conference on Learning Representations (ICLR 2022).

---

> ### Author Response · Authors · 2024-11-21
> **Response 3 for Reviewer q9DP**
>
> > Their results also leave a lot to be desired. The only environment where the method separates itself from existing methods is in the stochastic environments, but they only test in two such environments.
>
> Thank you for the feedback. To address your concern, **we have extended our experimental analysis by including two additional stochastic environments**: HIV Treatment [1] and Currency Exchange [2], two stochastic environments used in prior work[3]. We have added the results and descriptions of the environments in the updated paper draft. Please see the results below:
> | Env       | L-MAP ±        | TAP ±        | TT ±         | 1R2R ±       | CQL ±             | IQL ±        |
> |-----------|----------------|--------------|--------------|--------------|-------------------|--------------|
> | HIV       | 59.08 ± 1.96    | 54.95 ± 1.98 | 54.46 ± 3.30 | 56.45 ± 2.17 | **59.74 ± 1.11**  | 34.1 ± 1.2   |
> | Currency  | **106.78 ± 5.00** | 89.72 ± 3.90 | 79.28 ± 2.61 | 78.52 ± 2.08 | 93.96 ± 1.69      | 89.41 ± 2.83 |
>
> In the Currency Exchange environment, L-MAP significantly outperforms all other approaches, achieving a score of $106.78 \pm 5.00$, compared to the next best performer, CQL, which scores $93.96 \pm 1.69$. In the HIV Treatment environment, L-MAP performs on par with the best-performing baseline (i.e., it achieves a performance of $59.08 \pm 1.96$, which is comparable to the best-performing baseline, CQL, at $59.74 \pm 1.11$). These results highlight the effectiveness of our method in handling stochastic environments and underscore its versatility across different settings.
>
> **We also emphasize that although our algorithm primarily addresses stochastic dynamics, its benefits extend beyond environments with inherent stochasticity.** The dimensionality reduction necessary for handling high-dimensional state and action spaces introduces stochasticity through representation learning, even in deterministic settings. **Please see the enriched empirical analysis in the appendix, where we demonstrate this effect in response to questions asked by reviewer UcYT**. However, approaches such as TAP do not account for this stochasticity, and we show that our proposed approach outperforms TAP in most deterministic environments.
>
> [1]Ernst, D., Stan, G.-B., Goncalves, J., & Wehenkel, L. (2006). Clinical data based optimal STI strategies for HIV: A reinforcement learning approach. 45th IEEE Conference on Decision and Control
>
> [2]Bao, W., & Liu, X.-y. (2019). Multi-agent deep reinforcement learning for liquidation strategy analysis.
>
> [3]Rigter, M., Lacerda, B., & Hawes, N. (2023). One Risk to Rule Them All: A Risk-Sensitive Perspective on Model-Based Offline Reinforcement Learning. NeurIPS 2023.

---

> ### Author Response · Authors · 2024-11-23
> **Response 4 for Reviewer q9DP**
>
> We would like to thank you again for the valuable feedback. **We have now conducted additional ablation studies on progressive widening and parallel expansion and added the results to Figure 5 of the appendix.** Here is a summary of the ablation study results:
>
>
> Progressive Widening: Removing progressive widening drops mean normalized scores in high noise stochastic Mujoco environments from 68.7 to 54.77, demonstrating its crucial role in controlling state space expansion during the search.
>
>
> Parallel Expansion: Removing parallel expansion drops mean normalized scores in high noise stochastic Mujoco environments from 68.7 to 62.75, similar to using a shorter planning horizon, as already shown in our ablation study in Figure 5 of the Appendix. This confirms its importance in enabling efficient exploration within a fixed number of MCTS iterations, as parallel expansion allows the algorithm to evaluate multiple trajectories simultaneously.
>
>
> If you have any other concerns or feedback about the paper that might prevent you from increasing the score, please let us know, and we will address them. Thank you!

---

> ### Author Response · Authors · 2024-11-25
> **Follow-up**
>
> Dear reviewer q9DP,
>
> Thank you once again for your valuable feedback and dedicated service as a reviewer. We have thoroughly considered your suggestions and made substantial revisions to the manuscript in response. The discussion period will be closing soon on the **26th**. we truly appreciate your perspective and invite you to review our rebuttal along with the updated manuscript. We hope that our modifications address your concerns and provide the necessary clarity for you to reassess your evaluation. Please feel free to reach out if you need any further clarification or wish to discuss the changes in more detail.
>
> Sincerely,
>
> The Authors of paper #13144

---

> ### Author Response · Authors · 2024-11-27
> **Look forward to your reply**
>
> Dear Reviewer q9DP,
>
> Thank you again for your valuable feedback. As the revision period is approaching its end, we hope our responses have addressed your comments satisfactorily. Please let us know if you have any further questions or concerns.
>
> Best regards,
>
> The Authors of paper #13144

---

> > ### Comment · Reviewer_q9DP · 2024-12-01
> >
> > Based on your extensive additional evaluations and clarifications I have updated my score.

---

### Author Response · Authors · 2024-11-21
**Thank you to all reviewers**

We sincerely thank all the reviewers for their insightful comments and constructive suggestions. Below, we provide detailed responses to each reviewer. We have also submitted a revised version of our submission, with major text changes highlighted in blue. We look forward to further feedback and discussion.

---

### Meta-Review · Area_Chair_P4LH · 2024-12-21

**Metareview:**

This paper proposes an offline RL algorithm for high-dimensional continuous action spaces in stochastic environments. Action sequences are compressed into discrete codes via a VQ-VAE, a causal transformer predicts future codes, and planning is performed via MCTS. Experiments are performed on a few offline RL tasks, and ablation experiments are included as well.

Reviewers appreciated the motivation and intuition behind the proposed approach, and how the method can trade off latency with performance via the compute spent on planning.
Reviewers appreciated the comprehensive experiments (e.g., including both model-free and model-based baselines; including the Adroit hand task to show scalability; detailed ablations).

The proposed method contains several components and reviewers raised questions about which component contributes to the method's success, and whether the baseline comparisons were fair (e.g., the transformer baselines may not have utilized action compression); the reviewers recommended additional ablation experiments to address this concern. The reviewers also had concerns about the tasks used in the evaluation (why only two? how were these tasks made stochastic? why does the proposed method perform worse in deterministic settings?). Reviewers also had some suggestions to discussion additional related work.

Overall, the rebuttal period seems to have addressed the reviewers' initial concerns with the paper; reviewers all vote to accept the paper. Combined with the strengths of the paper noted above, I recommend that the paper be accepted.

**Additional Comments On Reviewer Discussion:**

The authors added an two new environments and additional ablation experiment to motivate their design decisions. The authors argued that the baseline comparisons were fair as the TAP baseline did include action compression, and they updated the paper to include full hyperparameters and analysis of heatmaps and stochasticity.

---

### Decision · Program_Chairs · 2025-01-22

Accept (Spotlight)